# Genome-Wide Identification and Expression Analysis Reveals the B3 Superfamily Involved in Embryogenesis and Hormone Responses in *Dimocarpus longan* Lour.

**DOI:** 10.3390/ijms25010127

**Published:** 2023-12-21

**Authors:** Mengjie Tang, Guanghui Zhao, Muhammad Awais, Xiaoli Gao, Wenyong Meng, Jindi Lin, Bianbian Zhao, Zhongxiong Lai, Yuling Lin, Yukun Chen

**Affiliations:** Institute of Horticultural Biotechnology, Fujian Agriculture and Forestry University, Fuzhou 350002, China; tangmengjie865@163.com (M.T.); zgh1209507033@163.com (G.Z.); awais9518@gmail.com (M.A.); gxl6957@163.com (X.G.); 18278493842@163.com (W.M.); 13616923127@163.com (J.L.); zbb@fafu.edu.cn (B.Z.); laizx01@163.com (Z.L.)

**Keywords:** *Dimocarpus longan*, somatic embryogenesis, B3 superfamily, expression analysis

## Abstract

B3 family transcription factors play an essential regulatory role in plant growth and development processes. This study performed a comprehensive analysis of the B3 family transcription factor in longan (*Dimocarpus longan* Lour.), and a total of 75 *DlB3* genes were identified. *DlB3* genes were unevenly distributed on the 15 chromosomes of longan. Based on the protein domain similarities and functional diversities, the DlB3 family was further clustered into four subgroups (ARF, RAV, LAV, and REM). Bioinformatics and comparative analyses of *B3* superfamily expression were conducted in different light and with different temperatures and tissues, and early somatic embryogenesis (SE) revealed its specific expression profile and potential biological functions during longan early SE. The qRT-PCR results indicated that *DlB3* family members played a crucial role in longan SE and zygotic embryo development. Exogenous treatments of 2,4-D (2,4-dichlorophenoxyacetic acid), NPA (N-1-naphthylphthalamic acid), and PP_333_ (paclobutrazol) could significantly inhibit the expression of the *DlB3* family. Supplementary ABA (abscisic acid), IAA (indole-3-acetic acid), and GA_3_ (gibberellin) suppressed the expressions of *DlLEC2*, *DlARF16*, *DlTEM1*, *DlVAL2*, and *DlREM40*, but *DlFUS3*, *DlARF5*, and *DlREM9* showed an opposite trend. Furthermore, subcellular localization indicated that DlLEC2 and DlFUS3 were located in the nucleus, suggesting that they played a role in the nucleus. Therefore, *DlB3s* might be involved in complex plant hormone signal transduction pathways during longan SE and zygotic embryo development.

## 1. Introduction

The B3 superfamily is a large plant-specific transcription factor, named for the B3 domain [1]. The first *B3* gene is the *viviparous·1* (*VPl*) gene, which has transcriptional activity [2], and its encode proteins have three domains (B1, B2, and B3 domains). The B3 domain is a highly conserved domain that can specifically bind to DNA [3,4]. The B3 domain is composed of 100–120 aa, including seven-stranded open β-barrel and two α-helices, forming a structure that binds DNA and acts by mosaicism with the DNA sulcus [5]. The *B3* superfamily plays an important regulatory role in plant growth [6,7]. According to the protein structure and functional characteristics, the B3 superfamily can be divided into four subfamilies: LAV (Leafy cotyledon2-Abscisic acid insensitive3-VAL) subfamily, ARF (auxin response factor) subfamily, RAV (related ABI3-VP1) subfamily, and reproduction meristem (REM) subfamily. The LAV family contains two subgroups, which are LEC2-ABI3 and VAL [7,8]. In *Arabidopsis thaliana*, except for the B3 structure, the structure of each subfamily is different. The ARF subfamily contains the auxin/IAA domain, the LAV subfamily contains the zf-CW domain, the RAV subfamily has the AP2 domain, and the REM subfamily only contains the B3 domain [8].

At present, the *B3* superfamily has been identified and analyzed in many plants, involving 118 *B3* genes in *Arabidopsis thaliana* [9], 91 *B3* genes in rice [1], 78 in *Phyllostachys edulis* [8], 97 in *Solanum lycopersicum* [10], 72 in *Citrus sinensis* [11], 69 in pummelo [11], and 57 in *Ananas comosus* L. [12]. The *B3* superfamily plays a vital role in plant embryonic development, growth, and stress resistance [13,14]. ARF transcription factors mediate auxin response by interacting with Aux/IAA transcription factors, thereby regulating the expression of auxin early response genes [7]. The ARF2 protein can bind to synthetic auxin response elements, which acted downstream of *HLS1* when affected by ethylene and light [15]. In *Arabidopsis thaliana*, AtMYB77 interacted with the AtARF7 protein, and this interaction led to a decrease in the number of lateral roots [16]. In addition, some *ARF* family members participated in the regulation of germ growth and seed development [17,18]. *AtARF6* promotes the maturation of flower organs and responds to stress [19]. Studies have shown that the overexpression of *SlRAV1* improved tolerance to diseases caused by fungi and bacteria, while silencing this gene enhances disease susceptibility [20]. The overexpression of *AtRAV1* accelerated leaf senescence [21], and it led to the delayed development of lateral roots and rosette leaves, while plants without *AtRAV1* expression flowered earlier, indicating that *RAV1* was involved in plant growth and development [22]. *AtVRN1* was the first gene in the *REM* family to be functionally defined which could maintain the response to vernalization in Arabidopsis [23,24].

The *LEC2* and *FUSCA3* (*FUS3*) genes are in the *LAV* subfamily, and *ABI3* also belongs to this subfamily; they are involved in the growth and development of seeds [25]. Yang et al. [26] found that the overexpression of *ABI3* could up-regulate the *LDP* gene in the absence of *FUS3*, thereby promoting oil accumulation. *FUS3* is a primary regulator of seed development, which promotes seed maturation and dormancy by regulating ABA/GA levels, and is also involved in the interaction of hormones in plants; it was highly expressed in the embryonic development stage [27]. In woody mangroves, the viviparous process included embryo formation and embryo germination. *FUS3* was expressed in the embryonic tissues and stimulated viviparous germination through an interaction with ABA, GA, BR, and auxin, indicating that it played an important role in the occurrence of the viviparous process [28]. Furthermore, *FUS3* regulated the downstream gene *AIL6* to play a role in seed dormancy and lipid metabolism [29]. The *LEC2* gene regulates the meristem and participates in the biosynthesis of plant hormones [7], and the increase in ABA inhibitors in the *fus3* mutant showed an intolerance to high temperature [27]. In the *fus3* mutant, *BBM* could not induce somatic embryogenesis, while in the *lec2* mutant, *BBM*-induced somatic embryogenesis was significantly reduced [30,31]. BBM transcription factors induced somatic embryogenesis by regulating the LEC1-ABI3-FUS3-LEC2 network [32]. *LEC2* regulates the transcriptional activity of *LEC1*, *L1L*, *ABI3*, and *FUS3* and plays an important role in maintaining the morphology of the suspensor, cotyledon development, synthesis of storage proteins during embryo maturation, and inhibition of premature seed germination [31,33,34,35,36]. In cassava, *MeLEC2* was highly expressed in somatic embryos, and the overexpression of *LEC2* could produce embryonic cells on the surface of mature leaves [37].

Longan (*Dimocarpus longan* Lour.) is a principal tropical/subtropical fruit tree. The development of the longan embryo is closely related to the yield and fruit quality [38]. However, the zygotic embryos of longan are encased in its pulp, making them difficult to obtain, and the development status of early embryos cannot be observed [39]. Therefore, it is essential to study the embryonic development of longan. Somatic embryogenesis (SE) is highly similar to embryonic development [40]. Since Lai et al. [41,42] established the high-frequency occurrence of the somatic embryogenesis system, it has provided a prominent experimental system for studying the SE of woody plants. In recent years, the second-generation and third-generation sequencing of the longan genome has provided complete and comprehensive genomic information for the related research on longan SE [43,44]. At the same time, the molecular mechanism of longan development was further revealed by improving the genetic transformation system of longan [45,46]. At present, the *B3* superfamily has been identified in several plants, while comprehensive data regarding the evolution and expression patterns of the *B3* superfamily in longan are still unavailable. Therefore, the longan *B3* superfamily was identified based on the longan genome database, and its expression patterns were analyzed under different exogenous hormone treatments, which laid a foundation for studying the regulatory mechanism of the *B3* superfamily during longan SE.

## 2. Results

### 2.1. Genome-Wide Identification of B3 Genes in Longan

To identify *B3* genes in the longan genome, the potential members of the *B3* superfamily were obtained by using the Hidden Markov Model. In total, 113 and 80 *DlB3s* were obtained from the second-generation and third-generation sequences of the longan genome database, respectively. The second-generation members were compared with the third-generation members, and SMART and Pfam databases were used to analyze the conserved domains of candidate sequences; the members that did not include the domain were removed (Dlo025515, Dlo025507). Due to the large difference in protein sequence length, it was impossible to compare, so the protein sequence which was shorter than 100 was deleted (Dlo024586). For the same coding sequences, we retained one of them: the coding sequence of *Dlo024135* was the same as *Dlo024091* and the coding sequence of *Dlo024139* was the same as *Dlo024094*, so we removed *Dlo024091* and *Dlo024094*. Finally, 75 *DlB3s* were obtained. According to the similarity of the *AtB3s* sequence and the classification of the phylogenetic tree, 75 *DlB3s* were named (Table 1).

The basic properties of *DlB3* genes, including protein length (aa), molecular weight (MW), pI, instability coefficient, and grand average of hydropathicity, are shown in Table 1. The length of the DlB3s ranged from 116 aa (DlREM33) to 1719 aa (DlREM42), with an average of 487.09 aa. The protein molecular weight ranged from 13.41 kD (DlREM33) to 192.57 kD (DlREM42), averaging 54.79 kD. The theoretical isoelectric points (pI) ranged from 4.50 to 10.85, and a total of 37 DlB3s were basic proteins. The instability coefficient was 8.92–73.08, and a total of 52 members were unstable proteins. The hydrophilicity was between −1.067 and 0.030, and only DlREM22 was a hydrophobic protein. In addition, subcellular localization analysis showed that DlB3 proteins were localized in different organelles: 62.67% of the DlB3 proteins were localized in the nucleus, 14.57% of the members were localized in the chloroplast, 12% of the members were localized in the cytoplasm, and four DlB3 proteins were localized in the plasma membrane. Furthermore, DlREM26/27 were localized in mitochondria, and DlREM10 and DlARF4 were localized in the peroxisome (Table 1).

### 2.2. Phylogenetic Relationship and Synteny Analysis of DlB3 Genes

A phylogenetic tree was constructed based on multiple sequence alignment between full-length protein sequences of 50 AtB3s and 75 DlB3s, using the ML method by MEGA5.05. According to the classification of *Arabidopsis*, DlB3s were mainly divided into four subgroups: ARF, RAV, LAV, and REM, with 18, 7, 8, and 42 members, respectively (Figure 1). The phylogenetic tree analysis showed that there were direct homologous gene pairs between longan and Arabidopsis: nine pairs of homologous genes in the ARF subfamily, four pairs of homologous genes in the LAV subfamily, and only one pair of direct homologous genes in the REM subfamily. Only DlREM4 had a separate branch, and most DlB3s have high similarity with *Arabidopsis thaliana*.

Chromosome localization of 75 *DlB3* genes in longan was performed. *DlB3* genes were unevenly distributed on the fifteen chromosomes of longan: three chromosomes (Chr3, Chr7, Chr9) contained one *DlB3* gene, thirty-four *DlB3* genes were located on Chr11, and DlREM40 and DlREM41 were located on the unknown chromosome (Figure 1). Chr11 showed tandem duplication, indicating hot spots for DlB3 gene distributions and gene duplication contributing to the amplification of the DlB3 family. To obtain collinearity gene pairs, TBtools software (V2.031) was utilized. The collinearity analysis of the *DlB3* family revealed 20 segmental duplication events in 34 members. These results suggested that tandem and segmental duplication may have been the main driving force of the evolution of the *DlB3* family.

### 2.3. Conserved Motifs, Structural Domains, and Gene Structure Analysis

To identify the conserved structure of longan B3 protein, 20 motifs were predicted using MEME software (https://meme-suite.org/meme/tools/meme (accessed on 5 September 2023)). The structural domains and gene structure of the DlB3 family were examined and visualized using the TBtools software. Motif 1 was detected in all DlB3 proteins. In the ARF subfamily, most members contained motif 2, motif 7, motif 8, motif 10, and motif 17. Only DlARF2 and DlARF4 had motif 1. In the LAV subfamily, except for DlVAL3-like, all included motif 14 and all contained motif 6 except DlLEC2, DlFUS3, and DlABI3. All members of the RAV subfamily contained motif 1, motif 6, and motif 14. The motif distribution of the REM subfamily was significantly different, and motif 14 was only contained in DlREM1, DlREM4, DlREM5, DlREM7, DlREM18, DlREM19, DlREM20 and DlREM21. In conclusion, they were placed in the same group as they probably have similar functions (Figure 2A).

Domain analysis of the DlB3 protein sequence revealed that all members contained B3 or B3_DNA domains (Figure 2B). In the ARF subfamily, most members contained AUX_IAA and auxin domains; DlAVL1, DlAVL2, and DlAVL3 of the LAV subfamily contained the zf-CW domain. In the RAV subfamily, DlTEM1, DlTEM2, DlRAV1, and DlRAV2 contained the AP2 domain.

To further understand the composition of *DlB3s*, their gene structures were compared. In total, 49.3% of *DlB3s* contained UTR, and 50% of the REM subfamily contained UTR; a few members of the REM and RAV subfamilies did not contain introns (Figure 2C).

### 2.4. Analysis Related to Cis-Acting Elements and Transcription Start Site in the DlB3 Promoters

To understand the potential function of the *DlB3* genes in various reactions, the 2 kb promoter region upstream of ATG was selected for cis-acting element analysis. These results suggest that *DlB3s* were associated with numerous phytohormone-related elements, including abscisic acid (ABA) response, gibberellin response, salicylic acid (SA) response, auxin response, and MeJA response (Figure 3A,B). With the exception of *DlREM7*, all *DlB3s* contained low-temperature-responsive elements, and 28 *DlB3s* contained defense- and stress-responsive elements. Additionally, nineteen *DlB3s* included circadian control cis-acting elements and cell cycle regulation cis-acting elements, and seed-specific regulation cis-acting elements were only present in the promoter of four *DlB3* genes. In summary, *DlB3s* may affect hormone regulation and various stress responses.

To further analyze the function of the 5′ terminal regulatory sequence of DlB3s, the intron and transcription start site of the 2000 bp sequence upstream of the DlB3 initiation codon ATG were predicted. The results of prediction analysis showed that only 5 of the 75 members had introns in the 5′ untranslated region. *DlARF1*, *DlARF6*, *DlVAL2*, *DlVAL2*-*like*, *DlREM1*, *DlREM3*, *DlREM9*, *DlREM16*, *DlREM21*, *DlREM26*, and *DlREM28* only contained one transcription start site, while the others all contained multiple transcription start sites. The number of transcription start sites of different members varied greatly. In general, there were four types of transcription start sites, A, T, C, and G, with A being the most common, followed by T and G (Appendix A).

### 2.5. Analysis of DlB3 Protein Interactions

To further explore the interaction relationships within the DlB3 family and among other family members, the PPIs were analyzed using STRING online software (https://cn.string-db.org/cgi/input?sessionId=btJ9BcX3f0Pk&input_page_show_search=on (accessed on 6 December 2023)). The results showed that DlB3s not only had strong interactions between family members but also interacted with other proteins. AUX1 interacted with most proteins (DlARF2, DlARF3, DlARF4, DlARF5, DlARF6, DlARF8, DlARF9, DlARF10, DlARF13, DlARF14, DlARF15, DlARF17, DlABI3). DlABI3 had the most interactions with eight proteins, including ABI4, ABI5, AGL15, AIP2, APRR1, AUX1, BZIP25, and BZIP8, followed by DlREM28, which interacted with six proteins. DlFUS3 interacted with five proteins (ABI4, ABI5, AGL15, BZIP8, CLF); DlLEC2 interacted with four proteins (ABI4, ABI5, AGL15, CLF). Therefore, it was speculated that DlB3s were functionally diverse, and the division of labor among subfamily members was different (Figure 4).

### 2.6. RNA-Seq Revealed the Expression Profiles of Longan DlB3 Genes in Different Tissues and Treatments

To further investigate the expression patterns of the *B3* superfamily in longan, the FPKM values of *DlB3s* were extracted from the longan transcriptome database. Of the 75 *DlB3* genes, 31 *DlB3* genes were not detected (Figure 5A). The remaining *DlB3s* displayed three expression patterns, with high expression at the NEC, GE, and EC to ICpEC stages. The expressions of *DlREM18*, *DlNGA1*, *DlNGA2*, *DlARF12*, and *DlTEM1* were high at the NEC stage, and they were less or even not expressed at the EC to GE stage. Twenty-two *DlB3s* had specific expressions at the GE stage, *DlFUS3* was not expressed at the NEC stage, and *DlFUS3*, *DlVAL3*, *DlARF5*, and *DlREM40* were expressed at a low level at the EC stage but increased gradually from the EC stage to GE stage, suggesting that these genes may significantly contribute to the early SE process. *DlLEC2* was not expressed at the NEC stage but remained stable from the EC stage to the GE stage. *DlARF7*, *DlARF10*, *DlARF14*, and *DlREM9* expression levels gradually decreased from the EC stage to the GE stage. It was demonstrated that these genes may promote the early SE process and maintain the embryonic state of EC.

Based on the transcriptome of longan in different tissues, RNA-seq data on the *DlB3* family were analyzed. A total of 47 *DlB3s* were detected (Figure 5B). This study revealed that 47 genes displayed tissue-specific expression in all tissues. The results showed that *DlARF5* and *DlNGA2* were specifically expressed in flower buds and flowers, but they were rarely or not detected in other tissues, revealing that they were mainly involved in floral organ development and flowering induction. Notably, *DlLEC2* and *DlFUS3* were highly expressed in the seed and might be involved in longan seed dormancy and germination. *DlREM19* and *DlREM21* were highly expressed in the root; *DlARF8*, *DlARF9*, *DlARF13*, *DlREM36*, and *DlNGA1* were highly expressed in the young fruit. These results suggested that *DlB3s* may be widely involved in the growth and development of longan, and the functions of multiple members were redundant.

There are two factors affecting longan SE, including light and temperature. According to the above, a series of core promoter elements were identified in the promoter sequences of *DlB3s*, which were involved in light and stress responsiveness. Hence, RNA-seq data were used to analyze the expression pattern of *DlB3s* under different treatments (Figure 5C). Most *DlB3s* responded to different light treatments, with higher expression under blue light. *DlARF14*, *DlARF16,* and *DlNGA2* were specifically expressed under white light treatment. *DlREM9/19/24/32*, *DlFUS3*, *DlARF17*, *DlNGA1*, and *DlTEM1* were highly expressed under dark conditions. It can be seen that the *DlB3* family also plays an important role in light response.

The expression patterns of *DlB3s* were further studied under different temperature conditions by analyzing the RNA-seq data for the longan EC; a total of 49 *DlB3s* were detected (Figure 5D). These results indicated the high expression levels of 26 *DlB3s* at 35 °C, and 13 *DlB3s* responded to low temperature (15 °C). In contrast, *DlFUS3*, *DlARF1*, *DlARF7*, and *DlREM15/19/32/34* were inhibited under high- or low-temperature treatment. In summary, the majority of the *DlB3* family may play a role in the self-repair process under temperature stress.

### 2.7. Expression Analysis of the DlB3 Family during Early SE

The analysis of the FPKM found that *DlARF5*, *DlARF16*, *DlTEM1*, *DlVAL2*, *DlLEC2*, *DlFUS3*, *DlREM9*, and *DlREM40* were significantly differentially expressed at the early stage. Therefore, the expression patterns of these genes were analyzed using qRT-PCR during longan early SE. The results showed that the qRT-PCR expression trends of *DlARF5, DlTEM1*, *DlVAL2*, *DlFUS3*, *DlREM9*, and *DlREM40* were similar to the RNA-seq (Figure 6B). The above results indicated that the expression of *DlB3s* had a particular spatial and temporal expression specificity, and the specific function needed to be further verified. All genes except *DlREM9* were highly expressed at the GE stage; it is concluded that they played a role in the GE stage; the transcription levels of *DlVAL2* and *DlREM40* were down-regulated from the EC stage to the ICpEC stage. *DlREM9* was explicitly expressed at the EC stage and up-regulated from the ICpEC stage to the GE stage; this study revealed that *DlREM9* significantly contributed to the induction and maintenance of longan EC. The expression of *DlFUS3* increased gradually from the EC stage to the GE stage, indicating that it had a positive regulatory effect in the SE of longan. *DlLEC2* was expressed from the EC to the GE stage and reached its peak value at the GE stage. These results suggested that the *DlB3* family was involved in longan early SE and played different roles. 

### 2.8. Analysis of the Expression Patterns of the DlB3 Family at Different Development Stages of Zygotic Embryos

The qRT-PCR experiments showed that the expression levels of *DlB3s* were divided into different patterns (Figure 7). *DlARF5* showed a gradual downward trend from S1 to S8; the expression level of *DlLEC2* was high throughout the whole stage of zygotic embryo development, which initially increased and then decreased, and the expression level was highest at the S4 stage and lowest at the S8 stage. The trend of *DlFUS3* and *DlARF16* fluctuated significantly, and *DlFUS3* showed a descending trend from the S5 to the S8 stage. *DlTEM1* and *DlREM9* exhibited opposite expression patterns, with *DlTEM1* decreasing from the S2 stage and increasing at the S8 stage, while *DlREM9* showed the opposite. *DlVAL2* maintained high expression levels, with the lowest expression at the S1 stage, while *DlREM40* maintained low expression levels and the highest expression at the S5 stage. Thus, *DlB3s* may serve a function in the development of longan zygotic embryos. 

### 2.9. Analysis of the Expression Patterns of the DlB3 Family under Different Exogenous Hormone Treatments

This study demonstrated that exogenous 2,4-D (2,4-Dichlorophenoxyacetic acid) treatment significantly inhibited the expression of *DlB3s* (Figure 8A). The qRT-PCR analysis of the *DlB3* family revealed that *DlARF5*, *DlFUS3*, and *DlREM9* were significantly higher than the control group under different concentrations of exogenous IAA treatment, *DlREM9* levels were significantly higher than those of the control, and the concentrations with the strongest promotion were 3 mg/L, 1.5 mg/L, and 0.5 mg/L (Figure 8B). However, under exogenous IAA treatment, the transcriptional levels of *DlARF6*, *DlTEM1*, *DlVAL2*, *DlLEC2*, and *DlREM40* were inhibited to different degrees. Under the treatment with auxin inhibitor NPA, the different concentrations of NPA treatment significantly reduced *DlB3* expression compared to the control group, indicating that exogenous NPA could significantly inhibit the expression of *DlB3s* (Figure 8C). Notably, the transcription levels of *DlARF5*, *DlFUS3*, and *DlREM9* showed a reverse trend compared with those of IAA treatment. However, the expression levels of *DlARF16*, *DlTEM1*, *DlVAL2*, *DlLEC2,* and *DlREM40* were consistent with a similar trend under IAA treatment, and their specific mechanisms still need to be further studied.

The treatment with 3, 6, 9, and 12 mg/L of ABA could promote the expression of *DlARF5*, *DlFUS3*, and *DlREM9*, demonstrating that exogenous ABA could promote the transcription (Figure 9A), whereas the expressions of *DlARF16*, *DlLEC2*, and *DlREM40* were suppressed under exogenous ABA treatment. Under 9 mg/L ABA treatment, the expression of *DlTEM1* was about 1.21-fold higher than that of the control, and *DlVAL2* was not significantly different between the 3 mg/L ABA treatment and the control.

The qRT-PCR result showed that under different concentrations of exogenous GA_3_ treatment, *DlARF16*, *DlTEM1*, *DlVAL2*, and *DlLEC2* were significantly lower than the control, suggesting that exogenous GA_3_ could inhibit their transcription levels (Figure 9B). Then, the transcription levels of *DlFUS3* and *DlREM9* were promoted to different degrees under GA_3_ treatment, and *DlFUS3* had the most obvious promotion effect under 12 mg/L GA_3_ treatment. The transcription levels of *DlARF5* and *DlREM40* were promoted only at partial concentrations. The qRT-PCR results showed that *DlB3* family expression was significantly lower under PP_333_ treatment concentrations, indicating that exogenous PP_333_ could significantly inhibit *DlB3* expression (Figure 9C). The expressions of *DlFUS3* and *DlREM9* showed a significant difference under PP_333_ treatment compared to exogenous GA_3_ treatment. To sum up, *DlB3s* may play a primary role in the regulation of the hormone signal transduction pathway in longan SE.

### 2.10. Subcellular Localization of DlLEC2 and DlFUS3

LEC2 and FUS3 are involved in plant embryogenesis. In order to further verify their potential functions in longan, their subcellular localization was verified. Based on the WoLF PSORT software (https://wolfpsort.hgc.jp/ (accessed on 3 October 2023)), DlLEC2 and DlFUS3 were predicted as nucleus localization proteins. Hence, the full-length coding sequences of DlLEC2 and DlFUS3 without a terminator codon (TGA/TAA) were fused with the EGFP (Enhanced Green Fluorescent Protein) and placed under the control of the 35S cauliflower mosaic virus (CaMV) promoter. The expression of pRI101-AN-35S::DlLEC2-EGFP and pRI101-AN-35S::DlFUS3-EGFP were observed with laser confocal microscopy using Agrobacterium infestation injected into onion epidermal cells, and 4’,6-diamidino-2-phenylindol (DAPI) was used for marking the nuclear localization. The results showed that onion epidermal cells transfected with the fusion expression vector containing the target fragments DlLEC2 and DlFUS3 showed green fluorescence mainly in the nucleus. In contrast, the fluorescence signal of the control group was distributed throughout the whole cell. After DAPI staining, pRI101-AN-35S::DlLEC2-EGFP, pRI101-AN-35S::DlFUS3-EGFP, and the positive control all showed fluorescence signals in the nucleus (Figure 10). Based on these results, DlLEC2 and DlFUS3 are nuclear transcription factors that play a role in regulating longan SE.

## 3. Discussion

### 3.1. DlB3 Family May Be Evolutionarily Conservative and Functionally Diverse

The *B3* family has multiple functions in plant growth and development and the abiotic stress response and participates in the hormone signaling pathways [1,47]. In this study, 75 *DlB3s* were identified; the lengths of the DlB3s ranged from 116 aa to 1719 aa. Tong et al. [48] found that the number of amino acids of 88 maize B3 proteins was between 105 and 1152 aa. In soybean, the number of amino acids of 145 B3 proteins was 72-1136 aa [14], and the number of amino acids of 97 B3 members of tomato was between 92 and 1317 aa [10]. Therefore, the number of amino acids represented in B3 proteins is relatively conserved in plants. In this study, motif 1 was detected in all B3 proteins, and most of the members contained motif 1 in maize. In *Gossypium hirsutum* [49], the distribution of *REM* family motifs was also specific, suggesting that conserved motifs play a specific role in the evolution of longan. In the process of plant evolution, it is speculated that genes without introns may be more conservative [50,51]. In this study, the *RAV* and *REM* subfamilies have genes without introns, and the genes of the same subfamily have a similar motif composition and intron number. Therefore, it is speculated that their evolutionary origin and molecular function are similar. Promoter cis-acting elements play an important role in the regulation of gene expression [52], and the prediction of promoter cis-acting elements lays a foundation for analyzing the function of *DlB3* genes. It was found that the promoter sequence of *DlB3s* contains many core elements, including stress response elements, hormone response elements, and light response elements. It is speculated that the *DlB3* family may play a role in the growth and stress response of longan through different cis-acting elements.

### 3.2. DlB3s May Be Involved in Longan Embryogenesis and Organ Morphogenesis

The overexpression of *AtLEC2* induced callus and somatic embryo formation in Arabidopsis [53], and *FUS3* also played key roles in controlling embryo development [54]. In this study, some longan *B3* genes were highly expressed from the EC stage to the GE stage and differentially expressed during the development of zygotic embryos. *DlFUS3* gradually increased from the EC to the GE stage, while *DlLEC2* was expressed from the EC to the GE stage, and both reached the peak in the GE stage, indicating that *DlLEC2* and *DlFUS3* might be involved in the process of longan SE. qRT-PCR is measured in the local region of the gene, and RNA-seq is measured in the full-length region of the gene [55,56,57]. The results of qRT-PCR in the early stage of longan SE were different from those of RNA-seq, presumably due to the fact that the samples were not from the same batch and the different detection regions. *DlARF5*, *DlLEC2*, and *DlFUS3* maintained low expression levels from the S5 stage to the S8 stage. However, *DlARF16* had high transcription levels from the S1 to the S8 stage. In summary, *DlB3s* played a role in the early embryo. *AtARF5* was involved in the formation of shoots from Arabidopsis calli [58], *CsARF19* was involved in the formation of the nucellar callus of citrus polyembryonic varieties, and *CsREM9* might be functional in early embryogenesis [11]. LEC2 and FUS3 recognize the CME at the FLC site and interact with each other in the vernalization process [59], and *ARF6* and *ARF8* were also involved in the flowering process [60]. In *Solanum lycopersicum*, *SlARF12* was highly expressed in the early stage of fruit development, indicating that it played a key role in fruit development and maturation [61]. RNA-seq indicated that 47 *DlB3s* showed tissue-specific expression at different levels in all tissues. *DlTEM2*, *DlARF5, DlARF7*, *DlREM38*, *DlREM9*, and *DlNGA2* were highly expressed in the flower bud stage, which suggested that these genes were involved in the flowering process of longan. Meanwhile, *DlREM24/29/32* were highly expressed in young fruit, and *DlLEC2* and *DlFUS3* were highly expressed in the seed. In summary, *DlB3s* are involved in the growth and development of longan plants.

### 3.3. DlB3s Were Involved in Longan SE through Hormones and Stress Response

Hormones play an essential role in regulating plant growth and development; auxin plays a vital role in the regulation of SE [62]. ABA can promote plant embryogenesis, seed dormancy, and fruit ripening, and the addition of an appropriate amount of ABA can promote the development of somatic embryos [63]. By mediating auxin biosynthesis and polar transport, ABA plays a role in the initiation of somatic embryogenesis by establishing auxin response patterns in the callus [64]. Abiotic stress has adverse effects on plant growth and development. Previous studies have shown that ABA, GA_3_, and auxin play a key role in plants under abiotic stress [65]. GA_3_ regulates the expression of transcription factors related to SE and participates in the process of SE [66]. It was found that *LEC2* reduced ABA levels and promoted somatic embryogenesis by controlling ABA8′-hydroxylase (*CYP707A1/2/3*) that catabolizes ABA [67,68]. In our study, exogenous GA_3_ could significantly inhibit the expression of most members of the *DlB3* family but up-regulate *DlFUS3* and *DlREM9* expression. Different concentrations of exogenous ABA promoted the transcription levels of *DlARF5*, *DlFUS3*, and *DlREM9* but inhibited the transcription of other *DlB3s*. 2,4-D plays a vital role in the regulation of somatic embryogenesis [69] In carrots, embryogenic cells differentiate into somatic embryos in hormone-free medium, while in medium containing 2,4-D, there was no differentiation. Therefore, the presence of 2,4-D inhibits the maturation of embryonic cells [70]. The effect of 2,4-D is phasic, promoting the induction of somatic embryos and inhibiting embryonic development [71]. In rice, *OsRAV9*, *OsRAV14*, and *OsRAV15* were down-regulated by exogenous IAA [72]. Studies have revealed that the expression of *LEC2* is closely related to the application of exogenous IAA. In explants cultured without IAA, the overexpression of the *LEC2* gene increased the content of endogenous IAA and promoted somatic embryogenesis, which proved that the *LEC2* gene might affect embryogenesis through the level of endogenous auxin [73,74]. In this study, supplementary exogenous 2,4-D and NPA significantly inhibited the expression of *DlB3s* in longan EC, and IAA induced the expression of *DlARF5*, *DlFUS3,* and *DlREM9*, while IAA inhibited other *DlB3s*; it could be seen that *DlB3s* were involved in the auxin signal transduction pathway. 

Appropriate stress can promote plant somatic embryogenesis [75,76]. It was found that sustained high temperature (34–38 °C) would lead to embryo development stagnation and small fruit formation [77], and the longan EC could not develop generally under 40 °C treatment [78]. The RAV family responded to various stresses [11,79]. Under different temperature treatments, the expression of the *DlB3* family suggested that most *DlB3s* were promoted at high temperatures (35 °C). The expression of *DlLEC2* was promoted under 35 °C treatment, while the expression of *DlFUS3* was inhibited under temperature stress. In addition, the transcription levels of 14 *DlB3s* were promoted under low-temperature treatment. Above all, consistent with the study of Ren et al. [14], *DlB3* genes may be involved in the response to abiotic stress and hormones during longan SE.

## 4. Materials and Methods

### 4.1. Plant Materials and Treatments

The ‘Hong He Zi’ (‘HHZ’) longan ECs, which involved the embryogenic callus (EC), incomplete compact pro-embryogenic cultures (ICpECs), and globular embryo (GE) were obtained as previously described by Lai et al. [41]. For exogenous hormone treatment, 2 g 18-day longan ECs were transferred to MS (Coolaber, Beijing, China) liquid medium (2% sucrose (Sinopharm, Beijing, China)) with 2,4-D (1.0 mg/L) (Yeasen, Shanghai, China) at 25 °C with 110 r·min^−1^ shaking in the dark for five days [80]. And then, the pre-cultured longan ECs were transferred to MS liquid basal medium (2% sucrose), supplemented with GA_3_ (3, 6, 9, and 12 mg/L) (Macklin, Shanghai, China), ABA (3, 6, 9, and 12 mg/L) (Yeasen, Shanghai, China), 2,4-D (0.5, 1.0, 1.5, and 2.0 mg/L), IAA (0.5, 1.0, 1.5, and 2.0 mg/L) (Coolaber, Beijing, China), N-1-Naphthylphthalamic acid (NPA: 5, 10, 20,30, 40, and 50 mg/L) (Coolaber, Beijing, China), and Paclobutrazol (PP333: 0.05, 0.1, 0.3, 1, 2, and 3 mg/L) (Solarbio, Beijing, China) with agitation at 120 rpm at 25 °C under dark conditions for 24 h, with three replicates [66]. Collection began in June when young fruits emerged from their cotyledon; the zygotic embryos of different developmental stages were collected from the cotyledon embryo stage of young fruit, and the zygotic embryos were collected every five days, which were labeled as S1, S2, S3, S4, S5, S6, S7, and S8 in turn [66,80]. All test materials were immediately frozen in liquid nitrogen and stored at −80 °C for subsequent tests.

### 4.2. Identification of the B3 Superfamily in Longan

The identification of the B3 superfamily was based on the second-generation sequence of longan from the NCBI Sequence Read Archive (SRA) database (SRR17675476) [43] and the third-generation sequence of longan from the NCBI database (PRJNA792504) [44]. The protein sequence of *Arabidopsis thaliana* was downloaded from https://www.arabidopsis.org/ (accessed on 4 September 2023) (Appendix A). The latest Hidden Markov Model (HMM) for the B3 superfamily (PF02363) (http://pfam.xfam.org/ (accessed on 15 July 2023)) was used, the potential members were obtained from the longan genome database with HMMER3.0, and the E-value was thresholded at 1 × 10^−5^. Then, all potential sequences were aligned using the DNAMAN software (V6 6.0.3.99) to eliminate repetitive proteins, combined with the results of NCBI analysis of the protein domain of the candidate sequence, and the members without the domain were removed. The molecular weight (MW), amino acid number (aa), isoelectric point (pI), average hydrophilicity (GRAVY), and instability coefficient were predicted using Expasy Protparam (https://web.expasy.org/protparam/ (accessed on 4 May 2022)). The subcellular localization of DlB3s was predicted with WOLF PSORT (https://wolfpsort.hgc.jp/ (accessed on 3 October 2023)), and members of the DlB3 family were named with reference to *Arabidopsis*.

### 4.3. Phylogenetic Evolution and Synteny Analysis of DlB3s

The protein sequences of 75 DlB3s and 50 AtB3s were aligned using the ClustalW by MEGA5.05. Then, the Maximum Likelihood (ML) method was used to construct the phylogenetic tree. The remaining parameters were designed: Bootstrap = 1000, Poisson model, and partial deletion (95%). The phylogenetic tree was perfected using the online software Chiplot (https://www.chiplot.online/ (accessed on 4 September 2023)). The diagrams of syntenic analysis were plotted using TBtools (V2.031, South China Agricultural University, Guangzhou, China) [81].

### 4.4. Analysis of Conserved Motifs, Structural Domains, and Gene Structure of DlB3s

The conserved motifs were predicted using the online software MEME (https://meme-suite.org/meme/tools/meme (accessed on 5 September 2023)) with the number of motifs set to 20, and the protein domain was predicted using NCBI (https://www.ncbi.nlm.nih.gov/Structure/bwrpsb/bwrpsb.cgi (accessed on 5 September 2023)). TBtools software was used to annotate the gene structure and visualize motifs and the conserved domains.

### 4.5. Analysis of Cis-Acting Elements and Transcription Start Site of DlB3s

A 2kb sequence upstream of the transcription start site of genes in the *DlB3* gene family was extracted from the longan genome file, and PlantCARE (http://bioinformatics.psb.ugent.be/webtools/plantcare/html (accessed on 5 September 2023)) was used to predict the cis-acting elements. Finally, TBtools was used for visualization. Then, the BDGP (https://fruitfly.org/seq_tools/promoter.html (accessed on 6 December 2023)) was used to predict the transcription start site, and the minimum promoter score was set to 0.8.

### 4.6. Protein Interaction Analysis of DlB3s

The protein sequences of DlB3s were selected, and the protein–protein interactions (PPIs) of DlB3s were analyzed with STRING (https://cn.string-db.org/cgi/input?sessionId=btJ9BcX3f0Pk&input_page_show_search=on (accessed on 6 December 2023)). *Arabidopsis Thaliana* was used as the model plant, and the required score was set at 0.700 to analyze the protein interactions of DlB3 members.

### 4.7. Expression Analysis of the DlB3 Family at the Early Stage of SE, Different Tissues, Different Light Quality, and Different Temperatures

Excel software (12.1.0.16120) was used to extract the FPKM values of *DlB3* family members from the transcriptomes of early SE (NEC, EC, ICpEC, and GE) (NCBI BioProject number: PRJNA891444) [43], different tissues (NCBI BioProject number: PRJNA326792) (young fruit, seed, flower, flower bud, leaf, pulp, root, and stem), different light qualities [82] (blue, white, dark as the control), and different temperatures (15 °C, 25 °C, and 35 °C) (NCBI BioProject number: PRJNA889670), normalized by log_2_^FPKM^. TBtools software was used to visualize and analyze the expression level of each member.

### 4.8. Subcellular Localization Analysis

The CDS sequences of *DlLEC2* and *DlFUS3* were selected to design subcellular localization primers by DNAMAN6 (Appendix A). The target gene sequence was amplified with PCR, the full-length coding sequences of *DlFUS3* and *DlLEC2* were inserted into pRI101-AN vector, and then the recombinant plasmid was transferred into Agrobacterium. pRI101-AN-35S::DlLEC2-EGFP and pRI101-AN-35S::DlFUS3-EGFP were transiently expressed in *Allium cepa*, whose epidermal cells were infiltrated by Agrobacterium. The onions were kept in a dark environment at 26 °C for three days; 4′,6-diamidino-2-phenylindol (DAPI) was used as a nuclear localization marker. The fluorescence signals of DlLEC2 and DlFUS3 proteins in cells were observed using an Olympus FV1200 confocal laser microscope (Tokyo, Japan), GFP wavelength 475 nm, and DAPI wavelength 450 nm.

### 4.9. RNA Extraction and qRT-PCR Analysis

Total RNA was extracted using a TransZolUp kit (TransGen Biotech, Beijing, China) for the material of the longan early SE (EC, ICpEC, GE), and the total RNA of different development stages of the zygotic embryo was extracted using a BioTeke kit (Cat#RP3301). The cDNA was carried out according to the instruction manual of Hifair^®^III 1st Strand cDNA Synthesis SuperMix for qPCR (gDNA digester plus) (Yeasen, Shanghai, China). Primer3 software (https://primer3.ut.ee/ (accessed on 9 September 2023))was used to design primers, and *DlACTB*, *DlEF-la*, and *DlUBQ* were used as internal control genes (Appendix A). The 20 μL reaction system contained the following: HRbioTM qPCR SYBR^®^Green Master Mix (No Rox) (Heruibio, Guangzhou, China), ddH_2_O 8.2 μL, 1 µL of 10-fold diluted cDNA, and 0.4 μL specific primer pairs. The operating parameters of the qRT-PCR were as follows: 95 °C for 30 s, followed by 40 cycles of 95 °C for 10 s, and 58 °C for 30 s. The relative expressions of *DlB3s* were calculated using the 2^−ΔCT^ method [83,84], and the data were imported into SPSS software (R26.0.0.0) to analyze significant differences; different letters representing significant differences were assessed with one-way ANOVA and Duncan test (*p* < 0.05); Graphpad 8.0.2 was used for the draft.

## 5. Conclusions

In this study, the genome of the longan *B3* superfamily was identified, and its expression in different stages of early somatic embryogenesis and exogenous hormones was analyzed. A total of 75 *DlB3* genes were identified in longan, and their bioinformatics and FPKM values in different transcriptomes were comprehensively analyzed, which revealed their specific expression profiles and potential biological functions during longan early SE. Subcellular localization indicated that DlLEC2 and DlFUS3 were located in the nucleus, suggesting that they played a role in the nucleus. Exogenous treatments with 2,4-D, NPA, and PP_333_ could significantly inhibit the expression of the *DlB3* family. Supplementary ABA, IAA, and GA_3_ suppressed the expressions of *DlLEC2*, *DlARF16*, *DlTEM1*, *DlVAL2*, and *DlREM40*, but *DlFUS3*, *DlARF5*, and *DlREM9* showed an opposite trend. The results showed that *DlB3* genes might be involved in the somatic embryo development and hormone response of longan, which could provide reference for the subsequent functional verification of the *B3* superfamily and the study of the hormone response mechanism.

## Figures and Tables

**Figure 1 ijms-25-00127-f001:**
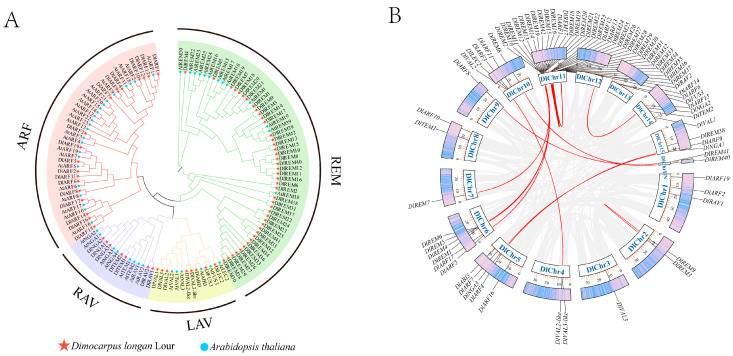
The phylogenetic tree of B3 members and collinear analysis of DlB3s. (**A**) Phylogenetic tree of longan (Dl) and *A. thaliana* (At) basic B3 proteins; (**B**) chromosome localization and collinear map of different *DlB3* genes in longan (gray line represents collinearity block in longan genome; red line represents linear gene pairs related to *DlB3* family genes).

**Figure 2 ijms-25-00127-f002:**
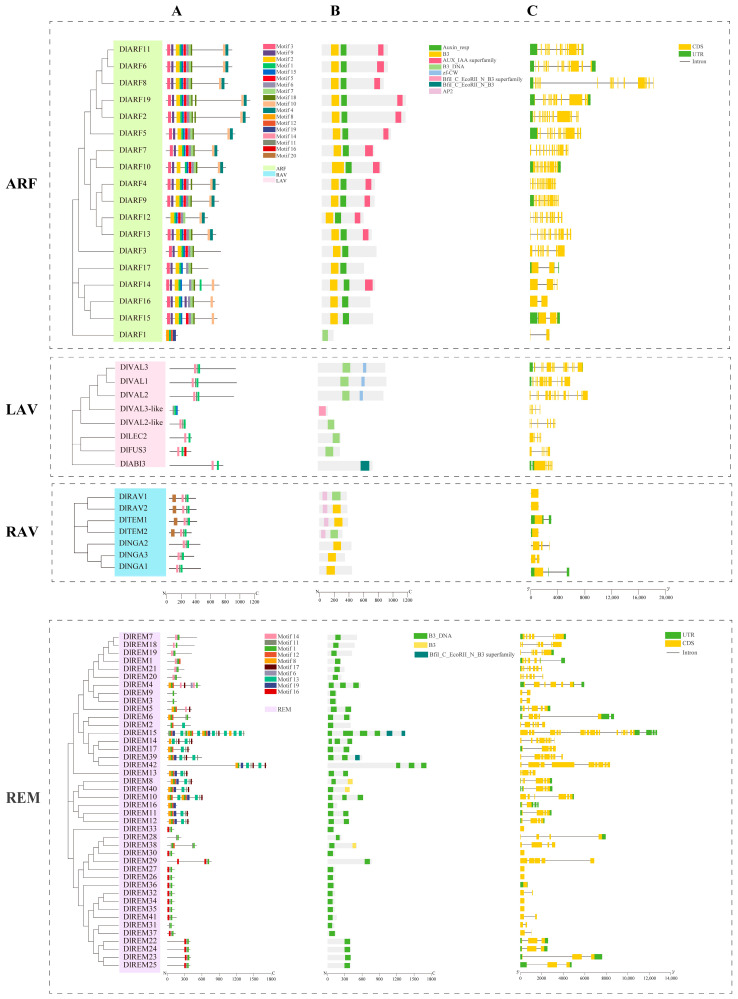
The motif organization, domain, and exon–intron structure of the *DlB3* family. (**A**) The motif organization of DlB3 proteins. Twenty conserved motifs predicted in B3 proteins are shown as differently colored boxes. (**B**) Domain of DlB3 proteins; different colors represent different domains. (**C**) Gene structure of *DlB3* genes. Green boxes indicate UTR; yellow boxes indicate exons; black lines indicate introns.

**Figure 3 ijms-25-00127-f003:**
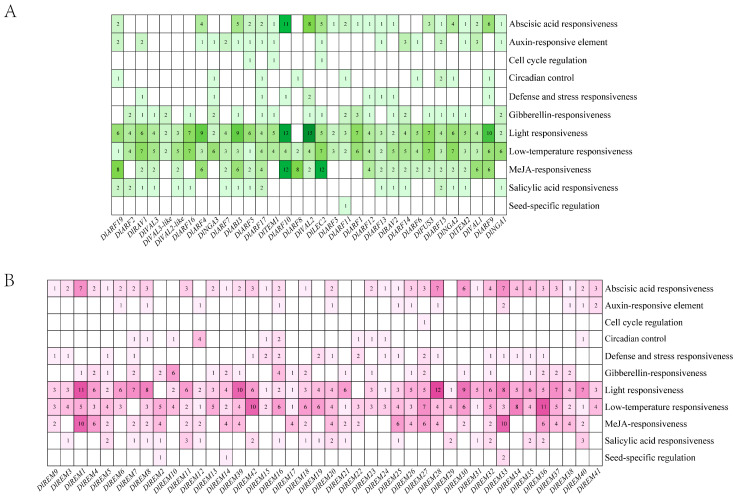
Distribution of cis-acting elements in promoters of *DlB3* longan. (**A**) Distribution of ARF, LAV, and RAV subfamily cis-acting elements in longan; (**B**) distribution of the REM subfamily cis-acting elements in longan. The blank spaces indicate no elements, green represents ARF, LAV, and RAV subfamily, purple represents REM subfamily, and color shades represent the number of components, and numbers in the figure indicate the number of elements.

**Figure 4 ijms-25-00127-f004:**
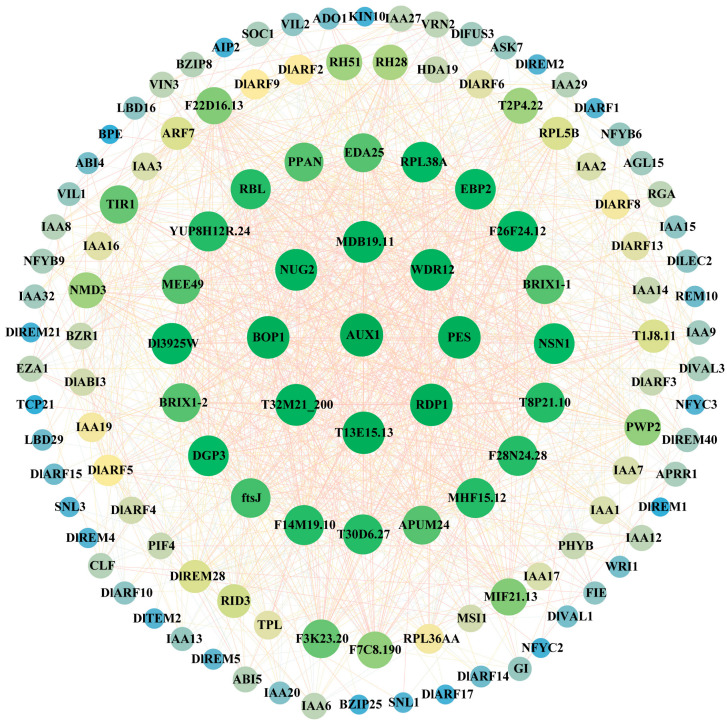
Protein interactions of DlB3 proteins in longan. Note: the connecting lines represent interactions between proteins; the size of the circle and the different color indicate the number of interacting proteins.

**Figure 5 ijms-25-00127-f005:**
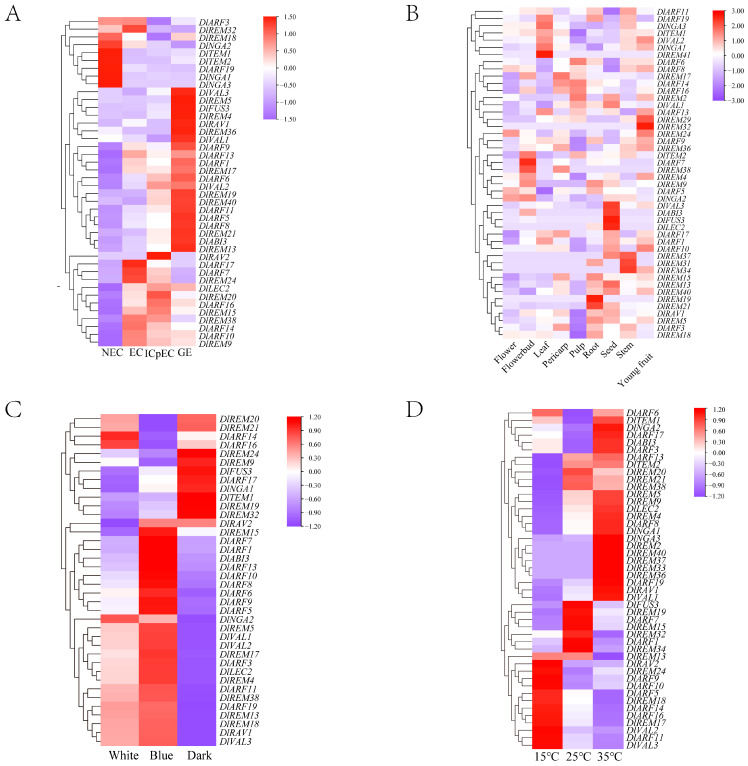
Expression patterns of *DlB3* family members based on FPKM values. (**A**) FPKM value of *DlB3* family during longan SE; (**B**) FPKM value of *DlB3* family in different tissues; (**C**) FPKM value of *DlB3* family under different light quality treatments; (**D**) FPKM value of *DlB3* family under different temperature treatments. The materials of different tissue were ‘SJM’ longan, and other materials were ‘HHZ’ longan. Different colors on the scale bar are log_2_^FPKM^, which represent different transcript levels.

**Figure 6 ijms-25-00127-f006:**
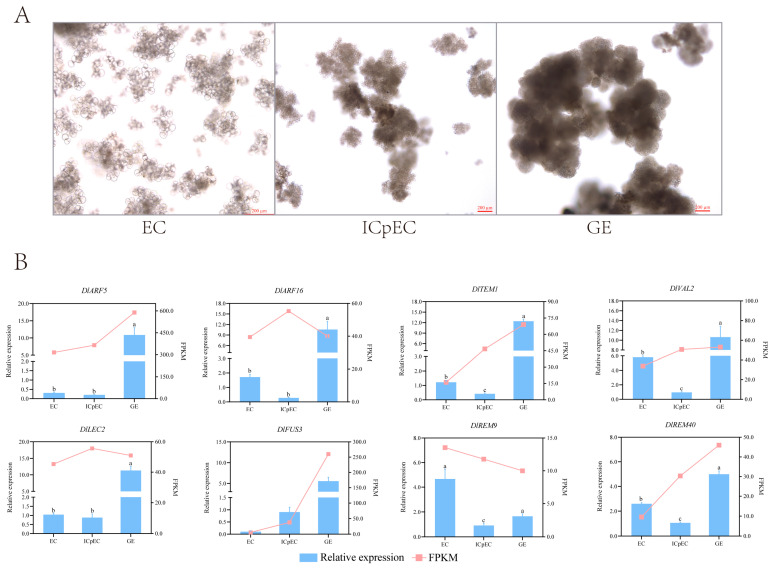
The early stage of longan somatic embryogenesis and eight *DlB3s* during the early somatic embryogenesis (SE) in longan with qRT-PCR. (**A**) Longan early stage of somatic embryogenesis; EC: embryogenic callus; ICpEC: incomplete compact pro-embryogenic cultures; GE: globular embryos. (**B**) The qRT-PCR analysis of *DlB3* genes the early stage of SE. Red line graph represents FPKM values; blue columns represent qRT-PCR results. Note: bars = 200 µm; the internal reference genes were *DlACTB*, *DlEF-la*, and *DlUBQ*, with three biological replicates, and significant differences are shown with lowercase letters a, b, and c, *p* < 0.05.

**Figure 7 ijms-25-00127-f007:**
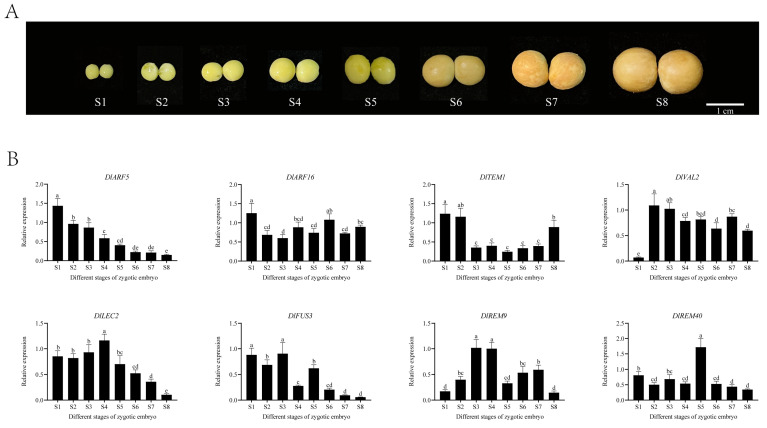
Expression patterns of longan *DlB3* family at different development stages of zygotic embryos. (**A**) The different developmental stages of zygotic embryos; (**B**) *DlB3* family relative expression of different stages of zygotic embryos. Note: the internal reference genes were *DlACTB*, *DlEF-la*, and *DlUBQ*, with three biological replicates, and significant differences are shown with lowercase letters a–e, *p* < 0.05.

**Figure 8 ijms-25-00127-f008:**
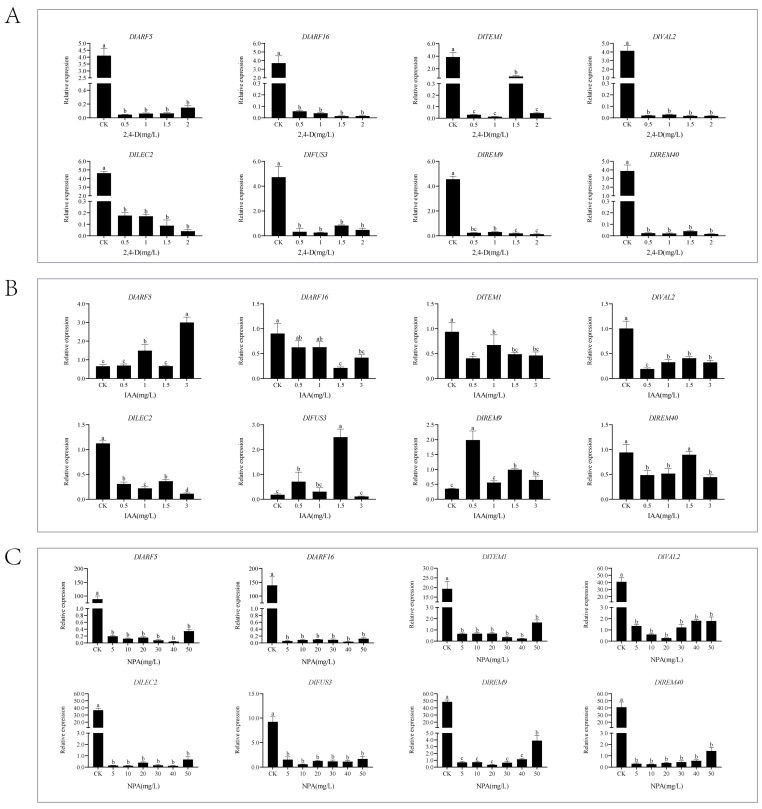
qRT-PCR analysis of the *DlB3* family under the 2,4-D, IAA, and NPA treatments. (**A**) qRT-PCR analysis of *DlB3* family under the 2,4-D treatment; (**B**) qRT-PCR analysis of *DlB3* family under the IAA treatment; (**C**) qRT-PCR analysis of *DlB3* family under the NPA treatment. Note: the internal reference genes were *DlACTB*, *DlEF-la*, and *DlUBQ*, with three biological replicates, and significant differences are shown with lowercase letters a–d, *p* < 0.05.

**Figure 9 ijms-25-00127-f009:**
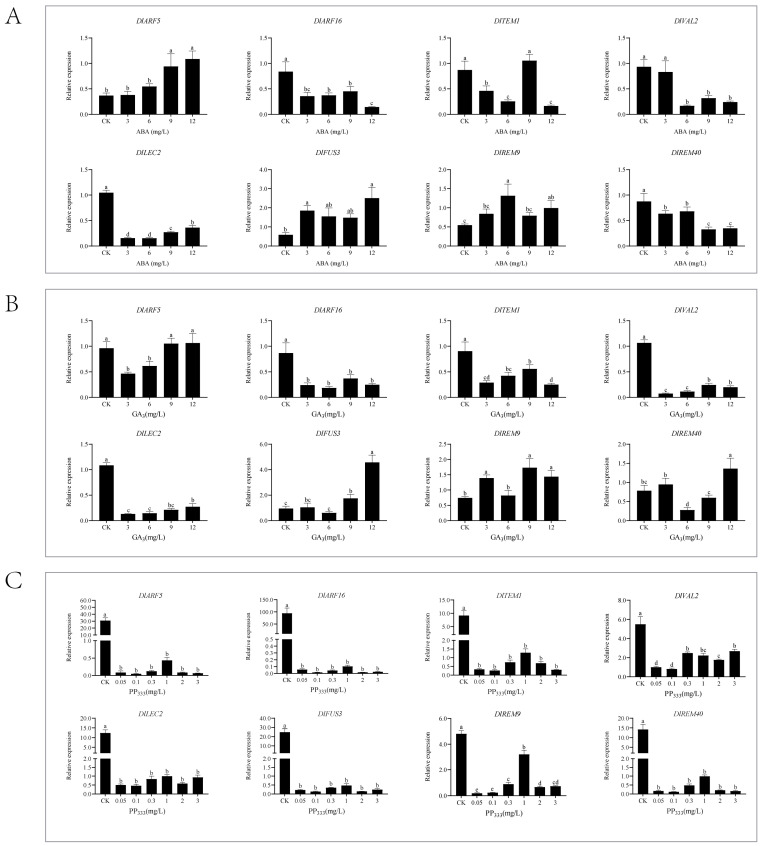
qRT-PCR analysis of the *DlB3* family under the GA_3_, ABA, and PP_333_ treatment. (**A**) qRT-PCR analysis of the *DlB3* family under the GA_3_ treatment; (**B**) qRT-PCR analysis of the *DlB3* family under the ABA treatment; (**C**) qRT-PCR analysis of the *DlB3* family under the PP_333_ treatment. Note: the internal reference genes were *DlACTB*, *DlEF-la*, and *DlUBQ*, with three biological replicates, and significant differences are shown with lowercase letters a–d, *p* < 0.05.

**Figure 10 ijms-25-00127-f010:**
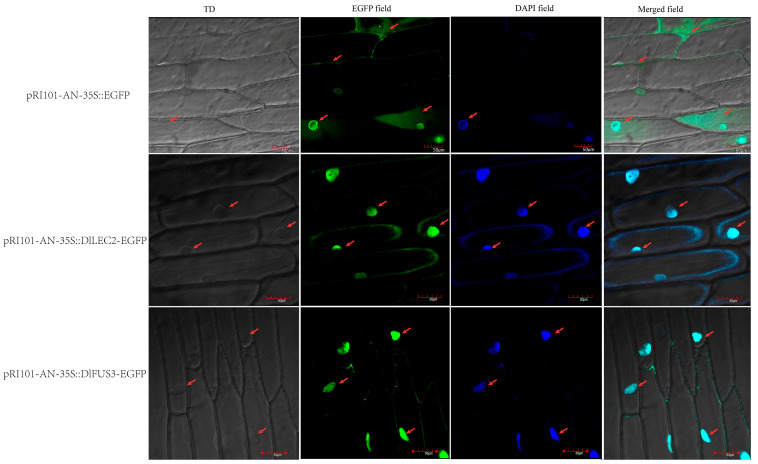
Subcellular localization of pRI101-AN-35S::EGFP empty, DlLEC2, and DlFUS3 in onion. Note: TD is a transmission light channel, the scale is 50 μM, and the arrows represent the localizations of the EGFP and DAPI fluorescence signal in cells.

**Table 1 ijms-25-00127-t001:** Physicochemical properties of longan *DlB3* family.

Gene ID	Gene Name	Number of Amino Acids	Molecule Weight/kD	pI	Instability Index	Grand Average of Hydropathicity	Subcellular Localization
Dlo023441	*DlARF1*	160	18,294.66	5.94	32.43	−0.490	Nucleus
Dlo001212	*DlARF2*	1141	126,748.28	6.28	63.42	−0.549	Nucleus
Dlo021672	*DlARF3*	745	80,871.67	6.61	53.59	−0.366	Nucleus
Dlo011752	*DlARF4*	723	80,752.32	6.63	52.54	−0.480	Nucleus
Dlo013412	*DlARF5*	942	104,338.60	5.27	53.55	−0.420	Nucleus
Dlo027446	*DlARF6*	900	99,373.37	5.89	64.53	−0.404	Nucleus
Dlo012202	*DlARF7*	715	79,496.01	8.08	54.59	−0.483	Nucleus
Dlo020967	*DlARF8*	844	94,263.26	6.09	58.42	−0.453	Nucleus
Dlo032463	*DlARF9*	719	80,129.41	6.51	58.48	−0.501	Nucleus
Dlo019051	*DlARF10*	815	91,040.33	6.72	51.55	−0.445	Peroxisome
Dlo022003	*DlARF11*	900	99,758.53	6.28	67.00	−0.448	Nucleus
Dlo024135	*DlARF12*	570	63,745.78	5.98	61.96	−0.564	Nucleus
Dlo024139	*DlARF13*	681	75,729.25	6.00	61.50	−0.536	Nucleus
Dlo026194	*DlARF14*	726	79,376.64	7.21	47.76	−0.300	Nucleus
Dlo029423	*DlARF15*	699	77,102.47	6.58	51.23	−0.378	Nucleus
Dlo011149	*DlARF16*	661	72,724.67	6.48	45.94	−0.432	Nucleus
Dlo013583	*DlARF17*	575	63,536.36	5.76	49.60	−0.350	Chloroplast
Dlo000294	*DlARF18*	1146	127,790.65	6.16	73.08	−0.668	Nucleus
Dlo030448	*DlVAL1*	909	99,933.47	7.27	51.27	−0.639	Nucleus
Dlo021533	*DlVAL2*	870	95,938.90	7.85	52.99	−0.702	Nucleus
Dlo008782	*DlVAL2-like*	220	24,623.14	6.52	29.32	−0.430	Nucleus
Dlo007002	*DlVAL3*	895	98,661.78	6.13	51.50	−0.658	Nucleus
Dlo008781	*DlVAL3-like*	131	15,276.82	9.14	19.37	−0.18	Cytoplasm
Dlo021632	*DlLEC2*	301	34,110.3	9.44	41.22	−0.662	Nucleus
Dlo027511	*DlFUS3*	292	32,766.84	5.91	36.95	−0.425	Nucleus
Dlo012591	*DlABI3*	726	81,352.54	6.41	59.52	−0.782	Nucleus
Dlo001570	*DlRAV1*	349	40,620.69	8.92	8.92	−0.778	Nucleus
Dlo025742	*DlRAV2*	357	40,982.28	8.74	39.47	−0.682	Nucleus
Dlo018238	*DlTEM1*	365	40,248.54	9.10	44.13	−0.523	Nucleus
Dlo030061	*DlTEM2*	293	33,555.67	6.83	48.96	−0.675	Nucleus
Dlo032565	*DlNGA1*	416	47,484.71	6.68	58.23	−0.862	Nucleus
Dlo029723	*DlNGA2*	408	46,538.25	6.55	58.48	−0.924	Nucleus
Dlo011844	*DlNGA3*	326	36,539.38	5.49	51.70	−0.798	Nucleus
Dlo014151	*DlREM1*	231	26,084.03	9.84	57.82	−0.490	Chloroplast
Dlo023420	*DlREM2*	403	46,286.78	6.34	60.32	−0.961	Chloroplast
Dlo004569	*DlREM3*	162	18,503.16	7.66	51.14	−0.531	Nucleus
Dlo014215	*DlREM4*	573	64,449.58	9.06	43.98	−0.388	Peroxisome
Dlo014216	*DlREM5*	422	47,100.77	8.71	38.46	−0.244	Cytoplasm
Dlo014217	*DlREM6*	402	46,348.13	9.77	45.32	−0.812	Nucleus
Dlo015857	*DlREM7*	510	57,348.01	6.24	49.93	−0.599	Nucleus
Dlo022556	*DlREM8*	440	50,988.45	8.88	34.69	−0.504	Cytoplasm
Dlo004420	*DlREM9*	162	18,559.08	5.93	53.58	−0.615	Nucleus
Dlo023431	*DlREM10*	616	71,420.96	9.07	41.80	−0.528	Chloroplast
Dlo023432	*DlREM11*	367	41,658.73	9.02	29.52	−0.526	Nucleus
Dlo023433	*DlREM12*	377	42,787.88	8.66	35.20	−0.503	Chloroplast
Dlo023434	*DlREM13*	358	41,439.44	5.43	43.25	−0.733	Nucleus
Dlo023435	*DlREM14*	439	49,915.2	8.36	43.19	−0.398	Cytoplasm
Dlo023438	*DlREM15*	1345	154,205.5	8.40	43.30	−0.475	Nucleus
Dlo023439	*DlREM16*	167	19,228.83	6.42	37.48	−0.517	Cytoplasm
Dlo023442	*DlREM17*	388	44,423.1	9.31	38.96	−0.570	Nucleus
Dlo023498	*DlREM18*	470	53,169.43	6.70	39.93	−0.530	Nucleus
Dlo023499	*DlREM19*	422	48,083.66	4.92	41.09	−0.443	Nucleus
Dlo023500	*DlREM20*	242	28,237.77	9.37	58.48	−1.067	Nucleus
Dlo023501	*DlREM21*	293	33,976.05	6.34	65.33	−0.879	Nucleus
Dlo024128	*DlREM22*	397	45,082.71	4.83	30.41	0.030	Plasma membrane
Dlo024132	*DlREM23*	406	45,600.43	4.50	32.55	−0.101	Plasma membrane
Dlo024182	*DlREM24*	397	44,966.33	4.72	27.44	−0.002	Plasma membrane
Dlo024185	*DlREM25*	395	44,420.04	4.54	33.52	−0.162	Plasma membrane
Dlo024514	*DlREM26*	124	14,588.86	10.04	41.36	−0.552	Mitochondrion
Dlo024515	*DlREM27*	125	14,499.78	9.93	40.53	−0.442	Mitochondrion
Dlo024518	*DlREM28*	240	26,941.38	9.58	38.23	−0.193	Nucleus
Dlo024524	*DlREM29*	767	88,189.41	9.31	41.63	−0.218	Chloroplast
Dlo024536	*DlREM30*	124	14,289.6	9.88	34.58	−0.440	Chloroplast
Dlo024545	*DlREM31*	120	13,677.86	9.45	43.74	−0.269	Nucleus
Dlo024552	*DlREM32*	131	15,368.62	10.85	37.20	−0.458	Cytoplasm
Dlo024555	*DlREM33*	116	13,414.94	10.00	31.16	−0.334	Cytoplasm
Dlo024557	*DlREM34*	124	14,469.44	9.62	41.69	−0.531	Chloroplast
Dlo024558	*DlREM35*	124	14,621.62	9.79	41.57	−0.563	Chloroplast
Dlo024600	*DlREM36*	123	14,476.87	10.50	32.73	−0.742	Cytoplasm
Dlo024601	*DlREM37*	142	16,080.3	7.05	56.77	−0.407	Chloroplast
Dlo031919	*DlREM38*	508	56,502.05	5.40	42.81	−0.358	Nucleus
Dlo023436	*DlREM39*	599	68,600.67	8.54	43.88	−0.525	Nucleus
Dlo032704	*DlREM40*	383	44,610.98	9.11	35.64	−0.606	Cytoplasm
Dlo033608	*DlREM41*	159	18,004.63	9.91	44.71	−0.561	Nucleus
Dlo023437	*DlREM42*	1719	192,573.79	9.24	44.13	−0.394	Chloroplast

## Data Availability

The original contributions presented in this study are included in the article material. Further inquiries can be directed to the corresponding authors.

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
