# Peer review of "Genome-Wide Identification and Expression Analysis Reveals the B3 Superfamily Involved in Embryogenesis and Hormone Responses in Dimocarpus longan Lour."

_ijms, 2023, doi:10.3390/ijms25010127_

Round 1

Reviewer 1 Report

Comments and Suggestions for Authors

Summary :

The authors reported the B3 family transcription factors in Dimocarpus longan. The manuscript contains interesting information.

General Comments :

The manuscript compares the promotors of all the B3 gene family transcription factors in this species and identifies the putative cis-element. However, the authors did not discuss the difference between promoters. Also, is there any conserved cis-acting of this gene family promoters? Please discuss this result.

In addition, there is no information regarding the promoter structure comparisons. Which region from the promoters that have 5 ’UTR’ introns of the promoters, TSS (transcription start site) location, etc ? The authors have the RNAseq data, I think this should be not a problem to annotate the 5 ’UTR’ introns of the promoters, TSS (transcription start site) location, etc. This is essential to elaborate on the difference between gene expression level and specificity of gene expression, for instance, tissue-specific expression or treatment-specific expression. I would suggest the authors to include this information.

Please also include the method for measuring FPKM from RNAseq in the material and method section. Since it appears in the results section.

Please provide and explain more regarding the construct of the plasmid, in the material and methods. I would recommend to provide this construct information in the supplementary.

Specific Comments :

Table 1.

Regarding the subcellular location, Plasma. Do you mean plasma membrane (cell membrane) ? I would suggest using this terminology to avoid confusion.

Figure 1A.

Please include the supplementary table regarding the Arabidopsis sequence used for creating the phylogenetic tree.

I found some “spaces” needed inside the manuscript. Please check again through the manuscript. For instance line 100, 356, 373, etc.

Line 169. Please revise the sentences and figure 2. "Yellow boxes indicate exons; Green boxes indicate UTR; Black lines indicate introns." There are some inconsistencies in the label color of the figure.

Line 171. It would be nice to change “2.4. Analysis related cis-acting elements in the DlB3 genes” into “2.4. Analysis related cis-acting elements in the DlB3 promoters.”

Please include a comparison graphic of RNAseq results from Figure 4A and qRT-PCR results Figure 5 since it is used the similar tissue used (EC, ICpEC and GE). Are the result of RNAseq similar with qRT-PCR ?.

Line 277-278. Please rewrite this sentence “significant differences with lowercase letters abc, P < 0.05”. This is due to the result in the figure 6 use “d” and “e” letter for significant difference group. Please apply this to other figures also.

It would be nice to state what kind of post-ANOVA test you used here in the figure legends ?. Even though it is already mentioned in the material and method section (line 474-475).

Line 408. Please elaborate “From June” in this sentences “From June, the zygotic embryos of different developmental stages were collected from the”

Line 428. The protein sequences of 75 DlB3s were aligned using the ClustalW by MEGA5.05. This is included also Arabidopsis sequence as presented in Figure 1 ? Check line 124.

I found many research questions that are still unknown regarding these B3 family transcription factors. I would recommend including the limitations or future research steps in the manuscript.

Comments on the Quality of English Language

Minor editing of English language required

Author Response

Response to Reviewer 1 Comments

Dear Reviewer:

Thank you very much for your comments. Those comments are all insightful and very helpful for revising and improving our manuscript, as well as the important guiding significance to our research. We have studied comments carefully and have revised the manuscript thoroughly, and the point-by-point responses to the comments are as follows.

All changes to the revised manuscript are indicated in the manuscript using track changes.

The manuscript compares the promotors of all the B3 gene family transcription factors in this species and identifies the putative cis-element. However, the authors did not discuss the difference between promoters. Also, is there any conserved cis-acting of this gene family promoters? Please discuss this result.

Authors response: Thank you so much for your comments. We have added this part in the manuscript. (Line 372-375)

In addition, there is no information regarding the promoter structure comparisons. Which region from the promoters that have 5 ’UTR’ introns of the promoters, TSS (transcription start site) location, etc ? The authors have the RNAseq data, I think this should be not a problem to annotate the 5 ’UTR’ introns of the promoters, TSS (transcription start site) location, etc. This is essential to elaborate on the difference between gene expression level and specificity of gene expression, for instance, tissue-specific expression or treatment-specific expression. I would suggest the authors to include this information.

Authors response: Thank you for your rigorous suggestions. We have revised those in the manuscript. (Line 185-192, 479-484)

Please also include the method for measuring FPKM from RNAseq in the material and method section. Since it appears in the results section.

Authors response: Thank you for your comments. We have added this part in the manuscript. (Line 495)

Please provide and explain more regarding the construct of the plasmid, in the material and methods. I would recommend to provide this construct information in the supplementary.

Authors response: Thank you for your comments. We have added this part in the manuscript. (Line 504-506)

Specific Comments :

Table 1.

Regarding the subcellular location, Plasma. Do you mean plasma membrane (cell membrane) ? I would suggest using this terminology to avoid confusion.

Authors response: Thank you for pointing out this problem in manuscript. We have corrected this mistake in the revised manuscript. (Line 116; Table 1)

Figure 1A.

Please include the supplementary table regarding the Arabidopsis sequence used for creating the phylogenetic tree.

Authors response: Thank you so much for your comments. We have added in the manuscript. (Supplementary Text S1)

I found some “spaces” needed inside the manuscript. Please check again through the manuscript. For instance line 100, 356, 373, etc.

Authors response: Thank you for your rigorous suggestions. We have revised those in the manuscript.

Line 169. Please revise the sentences and figure 2. "Yellow boxes indicate exons; Green boxes indicate UTR; Black lines indicate introns." There are some inconsistencies in the label color of the figure.

Authors response: Thank you for your rigorous comment. We have revised those in the manuscript. (Line 166-167)

Line 171. It would be nice to change “2.4. Analysis related cis-acting elements in the DlB3 genes” into “2.4. Analysis related cis-acting elements in the DlB3 promoters.”

Authors response: We gratefully appreciate for your valuable suggestion. We have revised those in the manuscript. (Line 169)

Please include a comparison graphic of RNAseq results from Figure 4A and qRT-PCR results Figure 5 since it is used the similar tissue used (EC, ICpEC and GE). Are the result of RNAseq similar with qRT-PCR ?.

Authors response: Thank you for your valuable and insightful comment. We have added those in the manuscript. In this study, we detected that some members showed similar trends, while some members showed different trends. It is speculated that the expression of DlB3s during SE has a certain degree of spatial and temporal specificity, and the specific function needs to be further verified. (Figure 5, Line 385-388)

Line 277-278. Please rewrite this sentence “significant differences with lowercase letters abc, P < 0.05”. This is due to the result in the figure 6 use “d” and “e” letter for significant difference group. Please apply this to other figures also.

Authors response: Thank you so much for your careful check. We have rewrite this sentence in the revised manuscript. (Line 291, 312, 335)

It would be nice to state what kind of post-ANOVA test you used here in the figure legends ?. Even though it is already mentioned in the material and method section (line 474-475).

Authors response: Thank you for your valuable and insightful comment. We have revised n the manuscript. (Line 524)

Line 408. Please elaborate “From June” in this sentences “From June, the zygotic embryos of different developmental stages were collected from the”

Authors response: Thank you so much for your careful check. We have rewrite this sentence in the revised manuscript. (Line 446-447)

Line 428. The protein sequences of 75 DlB3s were aligned using the ClustalW by MEGA5.05. This is included also Arabidopsis sequence as presented in Figure 1 ? Check line 124.

Authors response: Thank you for your valuable and insightful comment. We have revised n the manuscript. (Line 467)

I found many research questions that are still unknown regarding these B3 family transcription factors. I would recommend including the limitations or future research steps in the manuscript.

Authors response: Thank you for your valuable and insightful comment. We have revised n the manuscript. (Line 526-537)

Furthermore, all changes to the revised manuscript are indicated in the manuscript using track changes. Thank you so much.

Reviewer 2 Report

Comments and Suggestions for Authors

Good work and well written for the B3 superfamily analysis in Dimocarpus longan. I only have some minor suggestions.

1. the gene name would be italic in text, tables, and figures.

2. where is the transcriptome data early SE come from? The authors would supplemented the NCBI number or other database.

3. Why DlLEC2 and DlFUS3 were selected to subcellular localization analysis? It would be explain the reason in the section of methods or results.

Author Response

Response to Reviewer 2 Comments

Dear Reviewer:

Thank you very much for your comments. Those comments are all insightful and very helpful for revising and improving our manuscript, as well as the important guiding significance to our research. We have studied comments carefully and have revised the manuscript thoroughly, and the point-by-point responses to the comments are as follows.

All changes to the revised manuscript are indicated in the manuscript using track changes.

  1. the gene name would be italic in text, tables, and figures.

Authors response: Thank you for pointing out this problem in manuscript. We have corrected this mistake in the revised manuscript. (Figure 5)

  1. where is the transcriptome data early SE come from? The authors would supplemented the NCBI number or other database.

Authors response: Thank you for your rigorous suggestions. We have added this in the manuscript. (Line 496)

  1. Why DlLEC2 and DlFUS3 were selected to subcellular localization analysis? Itwould be explain the reason in the section of methods or results.

Authors response: Thank you so much for your comments. We have added in the manuscript. (Line 339-340)

Furthermore, all changes to the revised manuscript are indicated in the manuscript using track changes. Thank you so much.

Reviewer 3 Report

Comments and Suggestions for Authors

Dear authors, during November 21-24 2023, I have reviewed carefully the manuscript entitled:

Genome-wide identification and expression analysis reveals the B3 superfamily involved in embryogenesis and hormone responses in Dimocarpus longan Lour.

Authored by:

Mengjie Tang *, Guanghui Zhao *, Muhammad Awais, Xiaoli Gao, Wenyong Meng, Jindi Lin, Bianbian Zhao, 5 Zhongxiong Lai, Yuling Lin * and Yukun Chen *

Your manuscript requires several modifications to be accepted for publication.

Introduction:

Re-write this sentence:

Line 29-30

The earliest B3 gene was the viviparous·1 (VPl) gene, which had transcriptional activity [2], and its encoded protein had three domains (B1, B2, and B3 domains).

You mean that: The B3 domain was first identified in the VIVIPAROUS (VP1) gene from Zea mays, but not the earliest, or oldest.

Re-write this sentence.  Does LAV contain the zf-CW domain in Arabidopsis?.  Line 39-42

In Arabidopsis thaliana, except for the B3 structure, the structure of each subfamily were different. ARF contained the Auxin/IAA domain, LAV contained the zf-CW domain, RAV had the AP2 domain, and REM only contained the B3 domain [9].

The reference 15, does not correlate with the sentence. Line 49-51.

The ARF2 protein could bind to synthetic auxin response elements, which acted downstream of HLS1 when affected by ethylene and light [15].

Re-write this sentence: line 66-67

In woody mangroves, FUS3 was expressed in the process of viviparous, indicating that it played an important role in the occurrence of vi-viparous [24]. Please, indicate more precisely what is the viviparous process during mangrove growth and development.

Results:

Line 281 “The expression of the DlB3 family under different concentrations of 2,4-D treatment showed that exogenous 2,4-D could significantly inhibit the expression of DlB3s (Figure 281 7A)”.

Please, explain why 2,4-D inhibit the expression of the regulators of somatic embryogenesis (FUS3 and LEC2), and IAA, ABA and GA3.increased it.  

What is the role of 2,4-D during somatic embryo development?.

Add an excel list of all B3 genes analyzed in this work.

Discussion:

Complete with accurate references. Line 380-382. 2,4-D is the most important factor inducing somatic embryogenesis [51]. In rice, OsRAV9, OsRAV14, and OsRAV15 were down-regulated by exogenous IAA [52]. Not all plant species depend on 2,4-D to activate somatic embryogenesis, this is a fact.  You may know it is a herbicide and has been shown induced abnormalities during somatic embryogenesis process.

How many B3 family members genes have been demonstrated to be embryo lethal?. If so, what has been the molecular mechanism involved within mutants?

Explain how does LEC2 may be involved in the cell cycle regulation during somatic embryo development?

Taking all B3 genes from your work, what would it be the interaction of them?. What type of network will be developed?. And also what other type of genes will be required to make a full interaction between them?.

Conclusions:

You may have several conclusions about the role of the B3 superfamily genes in D. longan, please enlist at least 3 of them within the manuscript.

Literature:

Reference 7 is not well written, misspelled, I could not find the manuscript.

References 37 to 57 are out of place and does not correlate with the explanation in the discussion and part of material and methods.

Author Response

Response to Reviewer 3 Comments

Dear Reviewer:

Thank you very much for your comments. Those comments are all insightful and very helpful for revising and improving our manuscript, as well as the important guiding significance to our research. We have studied comments carefully and have revised the manuscript thoroughly, and the point-by-point responses to the comments are as follows.

All changes to the revised manuscript are indicated in the manuscript using track changes.

Dear authors, during November 21-24 2023, I have reviewed carefully the manuscript entitled:Genome-wide identification and expression analysis reveals the B3 superfamily involved in embryogenesis and hormone responses in Dimocarpus longan Lour.

Authored by:

Mengjie Tang *, Guanghui Zhao *, Muhammad Awais, Xiaoli Gao, Wenyong Meng, Jindi Lin, Bianbian Zhao, 5 Zhongxiong Lai, Yuling Lin * and Yukun Chen *

Your manuscript requires several modifications to be accepted for publication.

Introduction:

Re-write this sentence:

Line 29-30

The earliest B3 gene was the viviparous·1 (VPl) gene, which had transcriptional activity [2], and its encoded protein had three domains (B1, B2, and B3 domains).

You mean that: The B3 domain was first identified in the VIVIPAROUS (VP1) gene from Zea mays, but not the earliest, or oldest.

Authors response: Thank you so much for your comments. We have revised in the manuscript. (Line 30)

Re-write this sentence. Does LAV contain the zf-CWdomain in Arabidopsis?. Line 39-42

In Arabidopsis thaliana, except for the B3 structure, the structure of each subfamily were different. ARF contained the Auxin/IAA domain, LAV contained the zf-CW domain, RAV had the AP2 domain, and REM only contained the B3 domain [9].

Authors response: Thank you for your comments. We have rewrite this sentence  in the revised manuscript..(Line41-42)

The reference 15, does not correlate with the sentence. Line 49-51.

The ARF2 protein could bind to synthetic auxin response elements, which acted downstream of HLS1 when affected by ethylene and light [15].

Authors response: Thank you so much for your careful check. We have replaced this reference in the revised manuscript. (Line 50)

Re-write this sentence: line 66-67

In woody mangroves, FUS3 was expressed in the process of viviparous, indicating that it played an important role in the occurrence of vi-viparous [24]. Please, indicate more precisely what is the viviparous process during mangrove growth and development.

Authors response: Thank you for your comments. We have added this in the revised manuscript. (Line65-66)

Results:

Line 281 “The expression of the DlB3 family under different concentrations of 2,4-D treatment showed that exogenous 2,4-D could significantly inhibit the expression of DlB3s (Figure 281 7A)”.

Please, explain why 2,4-D inhibit the expression of the regulators of somatic embryogenesis (FUS3 and LEC2), and IAA, ABA and GA3.increased it.  

What is the role of 2,4-D during somatic embryo development?.

Authors response: Thank you for your comments. It was found 2,4-D inhibits the maturation of embryonic cells. FUS3 and LEC2 are genes for embryogenesis, so they are inhibited under exogenous 2,4-D treatment. Previous studies have shown that GA3 is involved in somatic embryogenesis, and regulates the expression of transcription factors related to somatic embryogenesis (Lai et al., 2002). And then by mediating auxin biosynthesis and polar transport, ABA plays a role in the initiation of somatic embryogenesis by establishing auxin response patterns in callus (Su et al., 2013), therefore, under the action of exogenous IAA and ABA were increased.

2,4-D is an important hormone that induces somatic cells in vitro culture of various plants to transform into embryonic cells, and different concentrations of 2,4-D can maintain longan at different somatic embryo stages.

Lai, Z., and Chen, C. (2002). . Changes of endogenous phytohormones in the process of somatic embryogenesis in longan. Chinese Journal of Tropical Crops (02), 41-47.

Su, Y.H.; Su, Y.X.; Liu, Y.; Zhang, X.S. Abscisic acid is required for somatic embryo initiation through mediating spatial auxin response in Arabidopsis. Plant Growth Regulation 2013, 69, 167-176.

Add an excel list of all B3 genes analyzed in this work.

Authors response: Thank you for your comments. 75 DlB3s identified in Table 1, used for qRT - PCR gene in Supplementary Table 1.

Discussion:

Complete with accurate references. Line 380-382. 2,4-D is the most important factor inducing somatic embryogenesis [51]. In rice, OsRAV9, OsRAV14, and OsRAV15 were down-regulated by exogenous IAA [52]. Not all plant species depend on 2,4-D to activate somatic embryogenesis, this is a fact.  You may know it is a herbicide and has been shown induced abnormalities during somatic embryogenesis process.

Authors response: Thank you so much for your comments. We have revised in the manuscript. (Line 415-419)

How many B3 family members genes have been demonstrated to be embryo lethal?. If so, what has been the molecular mechanism involved within mutants?

Authors response: Thank you for your comments. B3 family plays different regulatory roles in plant growth and development. At present, LEC2 and FUS3 genes promote embryogenesis, while VAL gene inhibits embryonic development.

Previous studies suggested that abi3 showed decreased sensitivity to abscisic acid. In Arabidopsis, the arf1 or arf2 mutant showed developmental delays, including flowering initiation and rosette leaf aging, while the double mutant arf7arf19 affected root growth and leaf expansion. fus3 mutants showed increased accumulation of abscisic acid inhibitors and showed intolerance to high temperatures. lec2 mutants exhibited excessive accumulation of anthocyanins in phyllocotyledon and cotyledon.

Explain how does LEC2 may be involved in the cell cycle regulation during somatic embryo development?

Authors response: Thank you for your comments. Studies have revealed that the expression of LEC2 is closely related to the application of exogenous IAA.In explants cultured without IAA, overexpression of LEC2 gene increased the content of endogenous IAA and promoted somatic embryogenesis, which proved that LEC2 gene might affect embryogenesis through the level of endogenous auxin. In our study, exogenous IAA significantly inhibited the expression of DlLEC2 in longan EC, and exogenous 2,4-D, ABA, GA3, NPA, PP333 also inhibited its expression. DlLEC2 was expressed from the EC to GE stage, and both reached the peak in the GE stage. It is speculated that DlLEC2 regulates somatic embryogenesis by participating in hormone signaling pathways, thereby participating in cell cycle regulation.

Taking all B3 genes from your work, what would it be the interaction of them?. What type of network will be developed?. And also what other type of genes will be required to make a full interaction between them?.

Authors response: Thank you so much for your comments. In our study, the analysis of FPKM found that DlARF5, DlARF16, DlTEM1, DlVAL2, DlLEC2, DlFUS3, DlREM9, and DlREM40 were significantly differentially expressed at the early stage. We speculate that the above genes interact with each other in the process of longan somatic embryogenesis, and the specific mechanism needs further study. And then, we have added the protein interaction analysis of DlB3s in the manuscript. (Line 194-203, 486-491; Figure 4)

Conclusions:

You may have several conclusions about the role of the B3 superfamily genes in D. longan, please enlist at least 3 of them within the manuscript.

Authors response: Thank you so much for your comments. We have added the conclusion in the manuscript. (Line 526-537)

Literature:

Reference 7 is not well written, misspelled, I could not find the manuscript.

References 37 to 57 are out of place and does not correlate with the explanation in the discussion and part of material and methods.

Authors response: Thank you so much for your careful check. Reference 7 was published on ‘Chemistry of Life’, is a Chinese article, DOI:10.13488/j.smhx.2013.03.018; References 37 to 57, confusion due to software issues. We have corrected this mistake in the revised manuscript. (Line 657-705)

Furthermore, all changes to the revised manuscript are indicated in the manuscript using track changes. Thank you so much.

Reviewer 4 Report

Comments and Suggestions for Authors

In this manuscript Tang and co-authors conduct a genome-wide characterization and a related expression analysis of B3 transciption factor superfamily in longan for which they find evidences for a role in embryogenesis and in signal transduction. Since among B3 members are included hormone responsive elements such as ARF, it is expected that they have a role in hormone-mediated signal transduction. In general, some assumptions are made with no solid supporting data other than gene expression (eg. lines 213, 221-222, 228, 249) that alone can only demonstrate, in my view, that it is expressed in that condition, not that it has a "critical" role.

Discussion could be improved: you performed different analyses so you have many data that can be used. You proposed possible functions in relation to  orthologs (how did you identify them? synteny, sequence similarity, ... ?): do they show similar characteristics/features with other species' counterparts or have peculiarities? For instance, did the results you get (like those regarding auxin supplement) match with expectations, also based on available literature? The presence of responsive elements in the promoter of such genes match their behaviour when such molecules are given? Can you find kinda of coexpression modules?

Getting into the text:

In order to obtain B3 genes, you used two different genome sequences, discarding proteins with the same sequence. Did this apply when comparing genomes or also within a single annotation? In the latter case, did you check whether the sequence of such genes and surrounding areas were actually identical (which may suggest assembly issues) or derive from recent tandem duplications?

Since the retained B3 genes is 75, how did you get there from 113? You did not explain that.

Protein length ranged from 116 to 1719 AA: did you check whether this range occur in other analyzed species or whether they could represent incomplete or fused proteins?

Did you check whether the conserved motifs identified by MEME actually represent known domains?

The analysis of RNAseq data in interesting and useful to hint at tissue- or time/condition-specific actions of the genes and especially those derived from treatments that could suggest specific responses of members of the B3 superfamily since you also analyzed the cis-acting elements in the promoters regions. Did you check whether such responses were in agreement with the related cis-acting elements you identified?

Section 2.5: please explain what the acronyms mean. Moreover, you report that only 44 DlB3s were expressed (lines 190-191), but then the number rises to 47 (line 204): how could this be? More in general, in the different panels of Fig. 4 you report different numbers of DlB3s, that I suppose represent those actually expressed in these tissues: you should indicate that. In panel 4D I count 46 genes, not 49 as you reported (line 224).

It is not clear to me what you want to indicate in line 225-226.

The genes you describe in lines 227-228 are probably just working in normal temperature conditions: have you hints about their putative function, also inferring that from orthologs that could reinforce your claim? Many genes can respond to temperature stresses and may have different roles. In any case "essential" (line 228) is not supported by any data, so please remove it.

Section 2.6. You cannot write that genes are "significantly differentially expressed" (line 239) if you do not provide any statistics about that.

Section 2.7. Please report what 2,4-D is.

Figure 4: values reported are log-transformed values, please correct. Please explain the acronyms in panel A.

Figure 7, 8: Why relative expression of controls (treated same way, theoretically) is so different between panels A, B, C? Did you use different housekeeping genes for normalization? You should explain it better. Please explain the acronyms, too.

Methods: how did you get expression levels? Did you map the reads yourself or you get them from elsewhere? In which form?

Comments on the Quality of English Language

You can use the present tense when describing the characteristics of B3 genes (introduction) since they still have.

Lines 100-105 are difficult to understand and sometimes in the text some phrases seem to be cut (ex.: lines 413-415): please read carefully the manuscript again.

Author Response

Response to Reviewer 4 Comments

Dear Reviewer:

Thank you very much for your comments. Those comments are all insightful and very helpful for revising and improving our manuscript, as well as the important guiding significance to our research. We have studied comments carefully and have revised the manuscript thoroughly, and the point-by-point responses to the comments are as follows.

All changes to the revised manuscript are indicated in the manuscript using track changes.

In this manuscript Tang and co-authors conduct a genome-wide characterization and a related expression analysis of B3 transciption factor superfamily in longan for which they find evidences for a role in embryogenesis and in signal transduction. Since among B3 members are included hormone responsive elements such as ARF, it is expected that they have a role in hormone-mediated signal transduction. In general, some assumptions are made with no solid supporting data other than gene expression (eg. lines 213, 221-222, 228, 249) that alone can only demonstrate, in my view, that it is expressed in that condition, not that it has a "critical" role.

Authors response: Thank you for your rigorous suggestions. We have revised those in the manuscript. (Line 236, 389, 405)

Discussion could be improved: you performed different analyses so you have many data that can be used. You proposed possible functions in relation to orthologs (how did you identify them? synteny, sequence similarity, ... ?): do they show similar characteristics/features with other species' counterparts or have peculiarities? For instance, did the results you get (like those regarding auxin supplement) match with expectations, also based on available literature? The presence of responsive elements in the promoter of such genes match their behaviour when such molecules are given? Can you find kinda of coexpression modules?

Authors response: Thank you for your comments. We proposed possible functions in relation to evolutionary relationship, We have added this part in the manuscript. (Line 373-376)

Getting into the text:

In order to obtain B3 genes, you used two different genome sequences, discarding proteins with the same sequence. Did this apply when comparing genomes or also within a single annotation? In the latter case, did you check whether the sequence of such genes and surrounding areas were actually identical (which may suggest assembly issues) or derive from recent tandem duplications?

Since the retained B3 genes is 75, how did you get there from 113? You did not explain that.

Authors response: Thank you for your rigorous comment. The second-generation longan genome is fragmented, however, the third-generation sequencing of the longan genome was complete and comprehensive genomic information. The three-dimensional structure of chromatin during early SE by Hi-C technique revealed the dynamic change of genome during longan SE. In the end, we chose the third-generation genomic data.

Protein length ranged from 116 to 1719 AA: did you check whether this range occur in other analyzed species or whether they could represent incomplete or fused proteins?

Authors response: Thank you for your comment. We have explained it in the discussion in the manuscript. (Line 362-366)

Did you check whether the conserved motifs identified by MEME actually represent known domains?

Authors response: Thank you for your comment. At the time of analysis, all motifs have been checked and all motifs represented known proteins.

The analysis of RNAseq data in interesting and useful to hint at tissue- or time/condition-specific actions of the genes and especially those derived from treatments that could suggest specific responses of members of the B3 superfamily since you also analyzed the cis-acting elements in the promoters regions. Did you check whether such responses were in agreement with the related cis-acting elements you identified?

Authors response: Thank you for your comment. RNA-seq revealed the expression profiles of longan DlB3 genes in different tissues and treatments, Analysis of promoter cis-acting elements showed that DlB3s contained a large number of hormone response elements, low temperature response elements, light response elements, etc. Compared with RNA-seq, it was found that some members responded to low temperature and different light quality. qRT-PCR analysis showed that most of the members were responsive to hormones, and in the process of somatic embryogenesis, endogenous hormones were in a dynamic form, and it was speculated that the members who did not respond were different in their action periods.

Section 2.5: please explain what the acronyms mean. Moreover, you report that only 44 DlB3s were expressed (lines 190-191), but then the number rises to 47 (line 204): how could this be? More in general, in the different panels of Fig. 4 you report different numbers of DlB3s, that I suppose represent those actually expressed in these tissues: you should indicate that. In panel 4D I count 46 genes, not 49 as you reported (line 224).

Authors response: Thank you so much for your comments. RNA-seq stands for RNA sequencing, we have added this in the Abbreviations. Because the transcriptome is different, the sample and treatment are also different, and the genes ultimately detected will be different, so the number of genes detected by each transcriptome is not the same.

It is not clear to me what you want to indicate in line 225-226.

Authors response: Thank you for your rigorous suggestions. We have revised those in the manuscript. (Line 239)

The genes you describe in lines 227-228 are probably just working in normal temperature conditions: have you hints about their putative function, also inferring that from orthologs that could reinforce your claim? Many genes can respond to temperature stresses and may have different roles. In any case "essential" (line 228) is not supported by any data, so please remove it.

Authors response: Thank you for your rigorous suggestions. We have revised those in the manuscript. (Line 241-242)

Section 2.6. You cannot write that genes are "significantly differentially expressed" (line 239) if you do not provide any statistics about that.

Authors response: Thank you for your valuable and insightful comment. The source of our transcriptomic data is presented in the material, and RNA-seq analyses are compared after homogenization.

Section 2.7. Please report what 2,4-D is.

Authors response: Thank you so much for your careful check. We have revised this in the manuscript. (Line 295)

Figure 4: values reported are log-transformed values, please correct. Please explain the acronyms in panel A.

Authors response: Thank you so much for your careful check. We have revised this in the manuscript, and the letters A-D represent expression patterns of four different transcriptomes . (Line 248)

Figure 7, 8: Why relative expression of controls (treated same way, theoretically) is so different between panels A, B, C? Did you use different housekeeping genes for normalization? You should explain it better. Please explain the acronyms, too.

Authors response: Thank you for your comments. The Ct value of the control of the same gene under different hormone treatments was consistent, but its expression level was calculated according to the expression level of different concentrations of the same hormone. Due to different hormones, the Ct value of each gene was also different, so the control was different in each group. The letters A-C represent the expression patterns of the DlB3 family under different exogenous hormone treatments.

Methods: how did you get expression levels? Did you map the reads yourself or you get them from elsewhere? In which form?

Authors response: Thank you for your comments. We use 2-ΔCT to calculate the expression. Regarding Livak et al. (Analysis of Relative Gene Expression Data using Real-Time Quantitative PCR. doi: 10.1006/meth.2001.1262. ), we substituted the Ct value of the internal reference gene into 2-ΔCT to obtain the 'Quality' values of the three internal reference genes, and used GEOMEAN algorithm in EXCEL to correct the ' Quality ' values of the three internal reference genes to obtain 'factor'. The 'Quality' value of the target gene was also calculated using 2-ΔCT. Finally, the 'factor' of the internal reference gene was divided by the 'Quality' value of the target gene, which were relative gene expression levels.

Comments on the Quality of English Language

You can use the present tense when describing the characteristics of B3 genes (introduction) since they still have.

Authors response: Thank you for your rigorous suggestions. We have revised those in the manuscript.

Lines 100-105 are difficult to understand and sometimes in the text some phrases seem to be cut (ex.: lines 413-415): please read carefully the manuscript again

Authors response: Thank you for your rigorous suggestions. We have revised those in the manuscript. (Line 101-102; 453)

Furthermore, all changes to the revised manuscript are indicated in the manuscript using track changes. Thank you so much.

Reviewer 5 Report

Comments and Suggestions for Authors

The manuscript titled "Genome-wide Identification and Expression Analysis Reveals the B3 Superfamily Involved in Embryogenesis and Hormone Responses in Dimocarpus longan Lour." requires major revisions.

The overall quality and resolution of Figures 5, 6, 7, and 8 are suboptimal, making it challenging to discern details. It is essential to enhance the clarity and resolution of these figures for improved visibility.

Provide a comprehensive conclusion summarizing the key findings of the study. Highlight the significance of the results in the context of embryogenesis and hormone responses in Dimocarpus longan Lour. Clearly articulate the implications and potential applications of the findings.

Establish a connection between the research results and fruit quality. Clearly articulate how the identified B3 superfamily genes are linked to aspects of fruit quality in Dimocarpus longan Lour. Elaborate on any potential implications for the improvement of fruit quality based on the study's outcomes.

Ensure consistency in using full names for abbreviations upon their first appearance. This will enhance clarity and facilitate better understanding for readers unfamiliar with the specific abbreviations used in the study.

Enhance the discussion section by minimizing the reliance on literature reviews. Focus on a more in-depth analysis and interpretation of the obtained results. Discuss the implications of the findings in the broader context of existing knowledge and highlight any novel aspects that contribute to the current understanding of the B3 superfamily in Dimocarpus longan Lour. Additionally, consider integrating potential avenues for future research based on the study's outcomes.

Author Response

Response to Reviewer 5 Comments

Dear Reviewer:

Thank you very much for your comments. Those comments are all insightful and very helpful for revising and improving our manuscript, as well as the important guiding significance to our research. We have studied comments carefully and have revised the manuscript thoroughly, and the point-by-point responses to the comments are as follows.

All changes to the revised manuscript are indicated in the manuscript using track changes.

The manuscript titled "Genome-wide Identification and Expression Analysis Reveals the B3 Superfamily Involved in Embryogenesis and Hormone Responses in Dimocarpus longan Lour." requires major revisions.

The overall quality and resolution of Figures 5, 6, 7, and 8 are suboptimal, making it challenging to discern details. It is essential to enhance the clarity and resolution of these figures for improved visibility.

Authors response: Thank you for your comments. We have replaced the above figures.

Provide a comprehensive conclusion summarizing the key findings of the study. Highlight the significance of the results in the context of embryogenesis and hormone responses in Dimocarpus longan Lour. Clearly articulate the implications and potential applications of the findings.

Authors response: Thank you so much for your comments. We have added the conclusion in the manuscript. (Line 526-537)

Establish a connection between the research results and fruit quality. Clearly articulate how the identified B3 superfamily genes are linked to aspects of fruit quality in Dimocarpus longan Lour. Elaborate on any potential implications for the improvement of fruit quality based on the study's outcomes.

Authors response: Thank you so much for your comments. RNA-seq analysis showed that B3 superfamily was involved in the growth and development of longan, and had regulatory effects on embryonic development and flowering. The growth and development of longan zygotic embryos are closely related to their production traits such as yield, seed size and fruit quality. In our study, DlB3s played a role in the early embryo and the growth and development of longan.Therefore, the use of molecular means to regulate the development of longan embryos provides a theoretical basis for solving the problems of longan yield and quality.

Ensure consistency in using full names for abbreviations upon their first appearance.

This will enhance clarity and facilitate better understanding for readers unfamiliar with the specific abbreviations used in the study.

Authors response: Thank you so much for your careful check. We have revised the abbreviations in the revised manuscript.

Enhance the discussion section by minimizing the reliance on literature reviews. Focus on a more in-depth analysis and interpretation of the obtained results. Discuss the implications of the findings in the broader context of existing knowledge and highlight any novel aspects that contribute to the current understanding of the B3 superfamily in Dimocarpus longan Lour. Additionally, consider integrating potential avenues for future research based on the study's outcomes.

Authors response: Thank you so much for your comments. We have revised in the revised manuscript. (Line 360-377; 410-411; 415-423; 527-537)

Furthermore, all changes to the revised manuscript are indicated in the manuscript using track changes. Thank you so much.

Round 2

Reviewer 1 Report

Comments and Suggestions for Authors

Thank you for considering my suggestion and comments. The manuscript is significantly improved compared to the last version. 

Author Response

Dear Reviewer:

Thank you for your comments, which have significantly raised the quality of the manuscript and have enabled us to promote the manuscript.

If you have any other suggestions for improving our manuscript, please do not hesitate to tell us, and we will rewrite or improve the manuscript by your suggestions. Thank you for your professional review.

Reviewer 3 Report

Comments and Suggestions for Authors

Dear authors and editors, I have reviewed the second round of the manuscript entitled:

Genome-wide identification and expression analysis reveals the B3 superfamily involved in embryogenesis and hormone responses in Dimocarpus longan Lour.

Authored by

Mengjie Tang*, Guanghui Zhao*, Muhammad Awais, Xiaoli Gao, Wenyong Meng, Jindi Lin, Bianbian Zhao, Zhongxiong Lai, Yuling Lin*, Yukun Chen*

The authors have made the pertinent corrections I have asked.

A few comments about this manuscript are enlisted.

Line 394: The reference 48 does not correspond to the written text. If so, is not Zhenhan instead?

Tong et al. [48] found that the number of amino acids of 88 maize B3 proteins

was between 105 and 1152 aa.

The reference 48:

Zhenhan, T.; Yanling, Z.; WANG Leili; Anran, L.; Wenqi, W.; Panpan, W.; Hongchang, L.; Qingqing, L.; Cuiling, W. Genome wide identification and expression pattern analysis of B3 gene family in maize. PRATACULTURAL SCIENCE 2023, 40, 2556-2570.

Line 446:

2,4-D just activates early stages of somatic embryogenesis, but is not the most important factor to induce somatic embryogenesis.  How can it be if inhibits embryonic development.

2,4-D is the most important factor inducing somatic embryogenesis [69] In carrots, embryogenic cells differentiate into somatic embryos in hormone-free medium; while in medium containing 2,4-D, there was no differentiation. Therefore, thepresence of 2,4-D inhibits the maturation of embryonic cells [70].[47]. Therefore, the role of 2,4-D has a stage, which promotes the induction of somatic embryos and inhibits embryonic development [71].

Line 455

It is not well redactated

Supplementary exogenous 2,4-D, NPA significantly inhibited the expression of DlB3s in longan EC, and IAA induced the expression of DlARF5, DlFUS3, and DlREM9, while IAA inhibited other DlB3s.

The references are well enlisted

Author Response

Response to Reviewer 3 Comments

Dear Reviewer:

Thank you very much for your comments. Those comments are all insightful and very helpful for revising and improving our manuscript, as well as the important guiding significance to our research. We have studied comments carefully and have revised the manuscript thoroughly, and the point-by-point responses to the comments are as follows.

All changes to the revised manuscript are indicated in the manuscript using track changes.

Dear authors and editors, I have reviewed the second round of the manuscript entitled:

Genome-wide identification and expression analysis reveals the B3 superfamily involved in embryogenesis and hormone responses in Dimocarpus longan Lour.

Authored by

Mengjie Tang*, Guanghui Zhao*, Muhammad Awais, Xiaoli Gao, Wenyong Meng, Jindi Lin, Bianbian Zhao, Zhongxiong Lai, Yuling Lin*, Yukun Chen*

The authors have made the pertinent corrections I have asked.

A few comments about this manuscript are enlisted.

Line 394: The reference 48 does not correspond to the written text. If so, is not Zhenhan instead?

Tong et al. [48] found that the number of amino acids of 88 maize B3 proteins

was between 105 and 1152 aa.

The reference 48:

Zhenhan, T.; Yanling, Z.; WANG Leili; Anran, L.; Wenqi, W.; Panpan, W.; Hongchang, L.; Qingqing, L.; Cuiling, W. Genome wide identification and expression pattern analysis of B3 gene family in maize. PRATACULTURAL SCIENCE 2023, 40, 2556-2570.

Authors response: Thank you so much for your comments. We have revised in the manuscript. (Line 670-671)

Tong, Z.; Zhang, Y.; Wang, L.; Li, A.; Wang, W.; Wang, P.; Liu, H.; Liu, Q.; Wang, C. Genome wide identification and expression pattern analysis of B3 gene family in maize. PRATACULTURAL SCIENCE 2023, 40, 2556-2570.

Line 446:

2,4-D just activates early stages of somatic embryogenesis, but is not the most important factor to induce somatic embryogenesis. How can it be if inhibits embryonic development.

2,4-D is the most important factor inducing somatic embryogenesis [69] In carrots, embryogenic cells differentiate into somatic embryos in hormone-free medium; while in medium containing 2,4-D, there was no differentiation. Therefore, thepresence of 2,4-D inhibits the maturation of embryonic cells [70].[47]. Therefore, the role of 2,4-D has a stage, which promotes the induction of somatic embryos and inhibits embryonic development [71].

Authors response: Thank you so much for your comments. We have revised in the manuscript. (Line 415)

In the study of carrot embryonic cells, it was found that embryonic cells differentiated in hormone-free medium, but did not differentiate in medium containing 2,4-D. The presence of 2,4-D destroyed the polarity distribution of endogenous auxin to a certain extent, thus inhibiting the further development and maturation of embryos.

Line 455

It is not well redactated

Supplementary exogenous 2,4-D, NPA significantly inhibited the expression of DlB3s in longan EC, and IAA induced the expression of DlARF5, DlFUS3, and DlREM9, while IAA inhibited other DlB3s.

The references are well enlisted

Authors response: Thank you so much for your comments. We have revised in the manuscript. (Line 423-426)

If you have any other suggestions for improving our manuscript, please do not hesitate to tell us, and we will rewrite or improve the manuscript by your suggestions. Thank you for your professional review.

Reviewer 5 Report

Comments and Suggestions for Authors

Accept in present form.

Author Response

(The authors gave the same response as above.)
